# Age mosaic of gut epithelial cells prevents aging

Peizhong Qin[1], Qi Wang[2], You Wu[1], Qiqi You[1], Mingyu Li[1] & Zheng Guo ●[1,3] ✉

Improving gut health by altering the activity of intestinal stem cells is thought to have the potential to reverse aging. The aged *Drosophila* midgut undergoes hyperplasia and barrier dysfunction. However, it is still unclear how to limit hyperplasia to extend lifespan. Here, we show that early midgut injury prevents the abrupt onset of aging hyperplasia and extends lifespan in flies. Daily transcriptome profiling and lineage tracing analysis show that the abrupt onset of aging hyperplasia is due to the collective turnover of developmentally generated "old" enterocytes (ECs). Early injury introduces new ECs into the old EC population, forming the epithelial age mosaic. Age mosaic avoids collective EC turnover and facilitates septate junction formation, thereby improving the epithelial barrier and extending lifespan. Furthermore, we found that intermittent time-restricted feeding benefits health by creating an EC age mosaic. Our findings suggest that age mosaic may become a therapeutic approach to reverse aging.

Aging is a complex process that is associated with the gradual decline of tissues homeostasis in the organisms[1,2]. The overall lifespan of animals correlates with the severity of degenerative changes in intestinal function[1,3]. Increasing evidence indicates that improve intestinal healthy has a profound impact on animal lifespan[1,4,5]. Therefore, the intestine has served as a target organ that carries out intracellular and extracellular aging interventions to extend lifespan and improve health[5–7].

Similar age-related structural and functional declines have been reported in the aging mammalian and *Drosophila* intestines[4,7–11]. The *Drosophila* midgut epithelium has emerged as a model system for the study of aging genetically, and contributes to the understanding of the mechanisms of mammalian intestinal aging[4,5,7]. The *Drosophila* midgut epithelium is a monolayer tissue composed of self-renewing intestinal stem cells (ISCs) that divide asymmetrically to give rise to enteroblasts (EBs) that differentiate into absorptive polypoid enterocytes (ECs) or enteroendocrine progenitor cells/enteroendocrine mother cells that give rise to a pair of enteroendocrine cells (EECs)[12–16]. In old flies, the midgut epithelium exhibits hyperplasia and barrier disruption, which associates with fly death[8,10,17–20]. How aging leads to midgut hyperplasia is still under investigation. It has been shown that age-related loss of midgut compartmentalization leads to commensal dysbiosis, which chronically produces ROS that activate JNK and PDGF/VEGF signaling[8,10,21,22] and disrupts NRF2 signaling[20,23]. These signals lead to ISC hyperplasia and the accumulation of mis-differentiated daughter cells that form multilayers overlying the basal membrane[8,10,20,22,24,25]. Concomitantly, altered localization of septate junction proteins and loss of tri-cellular junction (TCJ) components in aged ECs resulted in loss of intestinal barrier integrity[26–28]. And loss of gut integrity is associated with altered metabolic and immune signaling, which is tightly linked to age-related mortality[18,19,27,29]. If the hyperplasia in the aging midgut could be reversed, the establishment of an intact intestinal barrier would hold great promise for extending the lifespan.

In a healthy young *Drosophila* midgut epithelium, ISCs rarely divide. This results in a relatively slow turnover of the epithelium[8,10,17,30,31]. Upon injury, elevated JNK signaling in stressed or dying ECs induces Upd3 expression, which activates JAK/STAT signaling in ISCs[20,32–35]. Upd3 also activates *Vn* expression in visceral muscles, which then activate EGFR/Ras/MAPK signaling in ISCs[36–40]. JAK/STAT and EGFR signaling promote ISC proliferation and EB to EC differentiation to replace damaged ECs[36,37,40,41]. New EC generation alone is not sufficient; the formation of an intact intestinal epithelial

[1]Department of Medical Genetics, School of Basic Medicine, Institute for Brain Research, Tongji Medical College, Huazhong University of Science and Technology, Wuhan, China. [2]Department of Urology, Union Hospital, Tongji Medical College, Huazhong University of Science and Technology, Wuhan, China. [3]Cell Architecture Research Center, Huazhong University of Science and Technology, Wuhan, China. ✉e-mail: guozheng@hust.edu.cn

barrier requires the formation of an apical membrane initiation site (AIMS)/pre-assembled apical compartment (PAC) structures in pre-enterocyte between neighboring mature EC cells[42,43]. The injury repair process returns the midgut to a quiescent state within 3-5 days in young animals[20,21,32,33,36,44–47]. However, the physiologic status of the injured midgut during aging and the impact of early injury on longevity remain unknown.

Although adult ECs play a central role in nutrient absorption and tissue homeostasis, all ECs in the adult midgut when the fly eclosed from the pupa are generated by differentiation of adult midgut progenitor cells (AMPs)[48–51]. From 12 h after puparium formation, about 5000 AMPs differentiate into adult EC at the same time[15,50,52]. Moreover, pupal ISC only produces EECs, not ECs, during the pupal midgut development[15,48,53,54]. Therefore, the first wave of adult ECs is produced simultaneously during the pupal stage.

Here, we found that early midgut injury improves the barrier function of the aged midgut and extends lifespan by alleviating hyperplasia in aging flies. Our day-to-day examinations and transcriptome profiles showed that hyperplasia begins abruptly at around 20 days, at which time ECs collectively turnover due to ROS accumulation and Lamin degradation. Early injury promotes ISC proliferation and the formation of new ECs, which are embedded into the 'old' ECs, resulting in an epithelial age mosaic. This age mosaic prevents collective ECs death and facilitates the formation of septate junctions, thereby improving the epithelial barrier and extending lifespan. Additionally, we discovered that intermittent time-restricted feeding also benefits from the EC age mosaic. Taken together, our results show that the age mosaic of *Drosophila* midgut epithelial cells can reverse aging.

## Results

### Early injury improves the barrier function of the aged midgut and extends lifespan

In this study, we refer to intestinal injuries given to adult flies that eclosed within 4-6 days (d) as early injuries (EI). Since our studies were to investigate the effects of EI on aged flies, we reared mated female flies[55,56] by replacing fresh food every 2 days to avoid unexpected injury to the midgut during the aging process[30].

To examine the impact of EI on the lifespan of flies, we used the temperature-controlled EC driver *Myo1A-Gal4 tub-Gal80ts* (*Myo1Ats*) to temporally express the proapoptotic gene *Reaper* for 12 hours to induce EC loss at 4 d after eclosion[32], and then we recorded the survival rate of untreated (UT, control) and EI flies (Fig. 1a). Surprisingly, we found that EI flies had a significantly extended lifespan compared to UT flies (Fig. 1a), suggesting that EI is beneficial for fly health.

Disruption of barrier dysfunction is associated with increased mortality in aging flies[18,19,26]. To test whether EI improves barrier function in the aged midgut, we examined the integrity of the aging midgut barrier by feeding flies a non-absorbable blue dye (Smurf assay)[18]. The results show that the ratio of Smurf decreased in EI aged flies compared to the UT aged flies (Fig. 1b and Supplementary Fig. 1b), suggesting that EI improves the barrier integrity of the aging midgut.

We also induced EI by transiently feeding flies with bleomycin or paraquat for 24 hours at 4 d after eclosion. Both of bleomycin and paraquat feeding result in EC death and promote ISC proliferation[8,47,57]. Due to the acute toxicity of these chemicals, bleomycin and paraquat treatments did not extend the average lifespan of the fly population. However, in the surviving aging flies, we observed a significant improvement in the integrity of the gut barrier (Fig. 1c). Together, both early genetic EC ablation and chemical injury to the gut have a positive effect on the health of aged flies.

Frequent somatic mutations in *Notch* heterozygous mutant (*N^SSe11*/+) aged flies lead to formation of spontaneous *Notch* homozygous mutant EEC tumors (Pros+ cell cluster) in the midgut[58]. To test the effect of EI on tumor formation, we fed 4 d *N^SSe11*/+ flies bleomycin for

24 hours. Then, we counted the frequency and number of EEC tumors in the midguts of 35 d flies. Compared to UT flies, EI reduced the frequency of EEC tumors in the aged midguts (Fig. 1d, e). It has been demonstrated that the somatic mutation frequency is dependent on the ISC division rate[58], we therefore speculate that EI reduces ISC proliferation during fly aging.

### EI prevents the abruptly onset of hyperplasia in the aging midguts

We analyzed the effect of EI on ISC division rates on a day-by-day basis. After EI by *Myo1Ats>Reaper* or *Mex-Gal4 tub-Gal80ts* (*Mexts*) at 30 °C for 12 h, flies were reared at 18 °C (Fig. 1f and Supplementary Fig. 1a). With frequent fresh food replacement, the number of phospho-histone H3 staining positive (PH3+) cells in the UT midgut was maintained at a low level before 40 d (Fig. 1f). To our surprise, although the homeostatic midgut turnover is low, a significantly high number of PH3+ cells (hyperplasia) abruptly appeared at 45 d, and the hyperplasia persisted thereafter (Fig. 1f). By contrast, after an EI at 4 d, the number of PH3+ cells increased within 1 day and gradually decreased over the next 7 days, returning to a quiescent state similar to that of UT flies by 20 d (Fig. 1f). Notably, when the UT midgut entered hyperplasia at day 45, the EI midgut still maintained a low number of PH3+ cells (Fig. 1f). It remained in a state of hyperplasia alleviation even at 65 d (Fig. 1f).

We also transiently activated the JNK and JAK-STAT pathways in young ISCs (*esgts>Hep^act* or *esgts>hop^tuml*), which directly promotes ISC mitosis[8,32,37], to determine if early ISC proliferation also alleviates hyperplasia in aged flies. In both sets of experiments, the UT groups showed an abrupt onset of hyperplasia (Fig. 1g, h), whereas in EI group, the number of PH3+ cells in the midgut increased dramatically within one day after injury, gradually returned to a quiescent state and showed no hyperplasia even at 75 d (Fig. 1g, h). These experiments suggest that early ISC proliferation is sufficient to prevent the abrupt onset of hyperplasia.

At 25 °C, *esg-Gal4 > GFP* labeled ISC-EB pairs were dispersed throughout the young UT midgut, and the number of PH3+ cells was few (Fig. 1i–k). The hyperplasia occurred abruptly at around 20 days, characterized by the appearance of GFP+ cell clusters and a significant increase in the number of PH3+ cells (Fig. 1i–k). Following EI treatment by bleomycin or paraquat, the number of GFP+ cells increased rapidly, and within two days, there was a significant rise in the number of PH3+ cells (Fig. 1i–k). The number of PH3+ cells returned to a low level within 3-4 days (Fig. 1i–k), and the number and distribution of GFP+ cells in the midgut were similar to those in the UT group on day 12 (Fig. 1i). However, there was no hyperplasia in the EI midguts up to 30 d (Fig. 1i–k and Supplementary Fig. 1e). Our experiments suggest that EI prevents the onset of abrupt hyperplasia that occurs at about 20 d at 25 °C.

To exclude the effect of genetic background, we performed EI on *w^1118* flies and similarly found that EI alleviated hyperplasia in aged flies (Supplementary Fig. 1c, d). We also found that inducing injury on day 4 is not mandatory. Inducing transient damage by bleomycin or paraquat on day 14 before the onset of hyperplasia also alleviated the hyperplasia (Supplementary Fig. 1f). In addition, we found that upon EI, there was a slight but significant increase in gut diameter and area. However, after recovery from EI, both diameters and gut areas were restored to the UT group (Supplementary Fig. 1g–i). Taken together, our data suggest that EI or transient ISC proliferation may improve gut health and extend lifespan by alleviating the abrupt hyperplasia in the aged midgut.

### Hyperplasia alleviation is not due to compartmentalization or microbiota change

The hyperplasia in aged flies is closely linked to changes in the microbiota[19–22]. Recent studies suggest that bacterial dysbiosis is a consequence of age-related differentiation defects in acid-secreting

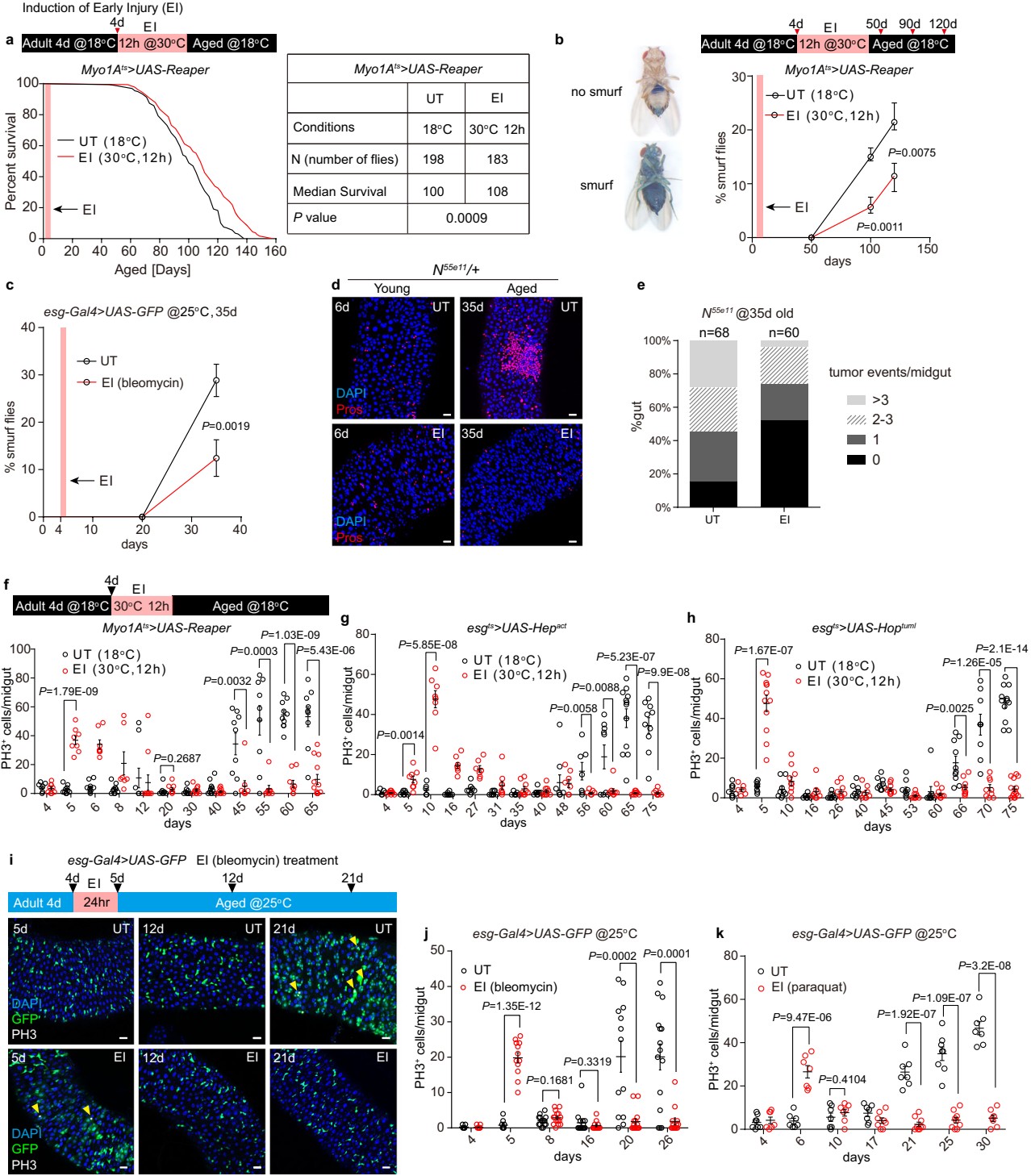

**Fig. 1 | Early injury extends lifespan and prevents the abruptly onset of hyperplasia. a** Survival analysis of *Myo1A$^{ts}$ > UAS-Reaper* flies reared at 18 °C (untreated, UT, n = 198) or transiently transferred to 30 °C for 12 h at 4 d to induce early injury (EI) (n = 183). The red stripe indicates the duration of EI. **b** Left; smurf assay for gut barrier integrity. Right; statistics of percentage of smurf flies in UT and EI group. n, number of flies. UT: n = 170, 166, 161. EI: n = 155, 145, 142. **c** Statistics of percentage of smurf flies in UT and EI (bleomycin feeding) group. UT: n = 110 (20 d), 94 (35 d). EI: n = 95 (20 d), 72 (35 d). **d** Representative images of Pros$^+$ tumor severity in young (6 d) and old (35 d) UT and EI *N$^{55e11}$/+* midguts. DNA staining (4′,6-diami-dino-2-phenylindole, DAPI) is blue in all images and Pros staining is red. **e** Frequency of Pros$^+$ tumor in 35 d UT and EI *N$^{55e11}$/+* midguts. n, number of midguts. **f–h** Time course of the statistics of phospho-Ser10-histone 3$^+$ (PH3$^+$) mitotic ISCs

per midgut in UT and EI flies. Flies were transferred to 30 °C for 12 h at 4 d to induce a transient EC ablation (*Myo1A$^{ts}$>Reaper*) (**f**) or a transient ISC proliferation (*esg$^{ts}$>Hep$^{act}$* and *esg$^{ts}$>Hop$^{tuml}$*) (**g**, **h**). n, numbers of midguts. The exact n numbers are provided in Source Data file. **i** Time course of representative images of posterior midguts stained with GFP and PH3 (white) in UT and EI (bleomycin) *esg-Gal4 > GFP* flies. Yellow arrowheads indicate the positions of PH3$^+$ cells. **j**, **k** Time courses of the statistics of PH3$^+$ mitotic ISCs per midgut in UT and chemically induced EI flies at 25 °C. Flies were fed either bleomycin (**j**) or paraquat (**k**) for 24 h at 4 d to induce EI. n, numbers of midguts. The exact n numbers are provided in Source Data file. Data are mean ± SEM. Significance was determined using two-tailed unpaired *t* test. Scale bars, 20 μm. Source data are provided as a Source Data file.

copper cells of the copper cell region (CCR)[22]. Therefore, we investigated whether EI improves midgut compartmentalization by counting Cut-stained copper cells[59,60]. However, we found that instead of improving compartmentalization of CCR, EI further reduced the number of Cut+ copper cells in aged midgut (Supplementary Fig. 2a, b). So we directly tested the effect of EI on midgut microbiota. Using 16 s rRNA sequencing and colony-forming units (CFU) counting, we found that the gut microbiota of EI flies was significantly altered compared to UT flies, specifically an increase of *Acetobacteria* (*Ab*) and a decrease of *Enterobacter* (*Eb*) in the EI midguts (Supplementary Fig. 2c, d). These results thus raised the possibility that either the enrichment of *Ab* or the depletion of *Eb* from the normal microbiome alleviate hyperplasia. Based on this assumption, we generated adult flies carrying a single commensal bacterium of *Ab* or *Eb* (mono-associated flies) (Supplementary Fig. 2e and see Methods). However, flies reared in *Ab* showed hyperplasia at 60 days old, and EI still alleviated hyperplasia of *Eb* reared flies, suggesting that changes in individual microbiota are not sufficient to alleviate hyperplasia in the aged midgut (Supplementary Fig. 2e).

To make the EI flies have the same microbiota as UT flies, we labeled EI and UT flies with different colors on their thorax and co-cultured them in the same cage (Supplementary Fig. 2f and see Methods). 16S rRNA sequencing and CFU counting revealed a similar microbiota profile between UT and EI flies (Supplementary Fig. 2g, h). However, even though EI fly has the same microbiota as UT fly, EI still alleviated hyperplasia in aged flies (Supplementary Fig. 2i). Moreover, we tracked cell proliferation in axenically reared EI and UT flies, and again, EI alleviated hyperplasia under axenic conditions (Supplementary Fig. 2j–l). Taken together, our results rule out that EI alleviates hyperplasia in aged flies by affecting the microbiota.

## Hyperplasia alleviation is not due to the ISC senescence

By defining *esg+ NRE*-lacZ− cells as ISCs[61], we found no significant change in the number of ISCs in the EI midgut at 21 d compared to the UT midgut (Supplementary Fig. 3a, b). We then asked whether EI or early ISC proliferation causes ISC senescence in aging flies. Our data showed that senescence markers, senescence-associated β-galactosidase (SA-β-gal) and HP-1[62] levels are not increased in EI ISCs of aged flies (Supplementary Fig. 3c–e), indicating that EI ISCs are not in a senescence state. To further demonstrate that EI ISCs could proliferate in response to the stimuli, we performed a second injury (SI) by *Erwinia carotovora carotovora 15* (*Ecc15*) feeding[47,63] at 21 d after the first EI by bleomycin or paraquat (Supplementary Fig. 3f). Oral infection with *Ecc15* induced a significant ISC proliferation in the EI aging midgut (Supplementary Fig. 3f), suggesting that EI ISCs are not senescent.

Furthermore, by using temperature-controlled EC or ISC drivers to induce EC death or JNK activation in ISC at an early age, and then shifting to 30 °C at 75 d to induce the second injury (Supplementary Fig. 3g), the EI midguts again showed ISC over-proliferation at this time point (Supplementary Fig. 3h, i), indicating that early ISC proliferation does not cause ISC to enter a senescent state.

## EI alleviates abrupt transcriptome changes

To elucidate the molecular mechanism by which EI alleviates abrupt hyperplasia, we performed mRNA sequencing of UT and EI midgut on a daily basis from 4 d to 28 d (Fig. 2a). Principal component analysis (PCA) showed that the transcriptomes of UT 4d-18d midgut were distinctly different from those of UT 19d-28d midgut, forming left and right clusters in PCA space (Fig. 2b), suggesting that the UT midgut undergoes drastic transcriptome changes during the 1-day period from 18 d to 19 d. However, in the EI group, except for the 6 d midgut that had just experienced injury, the EI transcriptomes did not undergo significant cluster splitting over time, and the majority of the EI 12d-18d transcriptomes were similar to that of UT 12d-18d, whereas the EI 19d-28d transcriptomes were spatially distinct from the UT 19d-

28d transcriptomes (Fig. 2b), suggesting that EI enabled the midgut to avoid abrupt transcriptome changes and switched the progression of aging.

Consistent with previous studies[64–68], gene set enrichment analysis (GSEA) revealed a significant downregulation of ribosomal genes in the midgut of UT 28 d flies compared to UT 4 d flies (Supplementary Fig. 3j), confirming the reliability of our sequencing results. We then performed Gene Ontology (GO) term enrichment analysis on differentially expressed genes (DEGs) between 28 d and 4 d UT transcriptomes. It was found that genes related to cell proliferation, migration, adhesion, and immune response were significantly upregulated in 28 d "old" midguts, while genes related to lipid and amino acid metabolism and digestion were significantly downregulated (Fig. 2c). Next, for the GO categories in Fig. 2c, we performed expression heatmap analysis for the top 10 genes of each GO category in the UT and EI transcriptomes, respectively, in chronological order from 4 d to 28 d (Fig. 2d). In the UT group, the heatmaps showed that the transcriptome changed abruptly toward aging in one day from 18 d to 19 d (Fig. 2d), but in the EI group, injury caused an upregulation of genes related to cell proliferation, division, and migration at 6 d, with a gradual decrease in expression during the following week. Notably, the EI transcriptomes did not change drastically at 19 d, and after 22 d, the EI transcriptome showed a trend toward aging, but was considerably milder than the aging state of the UT group (Fig. 2d). To further demonstrate at the whole-transcriptome level that aging changes abruptly and EI alleviates this tendency, we performed temporal trend analyses of all genes in the UT and EI groups and identified five clusters of gene trajectories that change with age, respectively (Fig. 2e and Supplementary Fig. 3k, l). While in UT clusters 2 and 3, the expression trajectories of 6142 (up) and 930 (down) genes were abruptly changed at 19 d (Fig. 2e and Supplementary Fig. 3k), the expression trajectories of EI clusters showed no abrupt alteration at 19 d (Fig. 2e and Supplementary Fig. 3l). Taken together, our results confirm that the aging process of the midgut is not linear, but rather occurs abruptly at an aging time point, and that EI alleviates this abrupt change and slows aging.

Among the GO categories abruptly upregulated at 19 d in the UT transcriptome, we found an abrupt up-regulation from 18 d to 19 d in genes related to JNK (*Ets21C, mkp3, puc, Diap1*), JAK-STAT (*upd3, upd2, upd1, dome, socs36E, hop, Sox21a*), EGFR (*vn, rho, Krn, spi, CycE*), and IMD/Relish (*PGRP-LE, Rel, DptA, DptB*) signaling pathways, which are involved in EC replacement and facilitation of ISC division (Fig. 2f)[8,10,17,20,21,25,32,37,38,40,69–76]. However, in the EI transcriptomes, the above-mentioned genes did not undergo abrupt up-regulation at 19 d and remained mildly expressed even at 28 d (Fig. 2f). The temporal expression pattern of these genes suggests that EI treatment enabled the midgut not to undergo an abrupt, dramatic EC turnover at the time when aging should have occurred (19 d), resulting in alleviation of aging.

## EI slows down EC turnover in aged midgut

To verify that the expression of signaling pathways associated with EC turnover was significantly reduced in the EI midgut, we examined the expression of *upd3-LacZ*, which reflects the JAK-STAT ligand expression in dying ECs[32], *10XSTAT*-GFP, which reflects the activation of JAK-STAT signaling[32], and *vn-LacZ*, which reflects the EGF ligand expression in visceral muscle cells[37], in 25 d UT and EI midguts. The results showed that the expression levels of all three markers were significantly reduced in the aged EI midgut compared to the strong activation in the aged UT midgut (Fig. 3a–d), confirming our transcriptome results. Moreover, our reverse transcriptase quantitative PCR (RT-qPCR) results also confirmed that the mRNA transcript levels of *upd3*, *socs36E*, and *vn* were significantly reduced in the aged EI midgut (Fig. 3e–g). Collectively, our results indicated that the EC of aged EI midgut had a significantly lower turnover rate compared to the UT group.

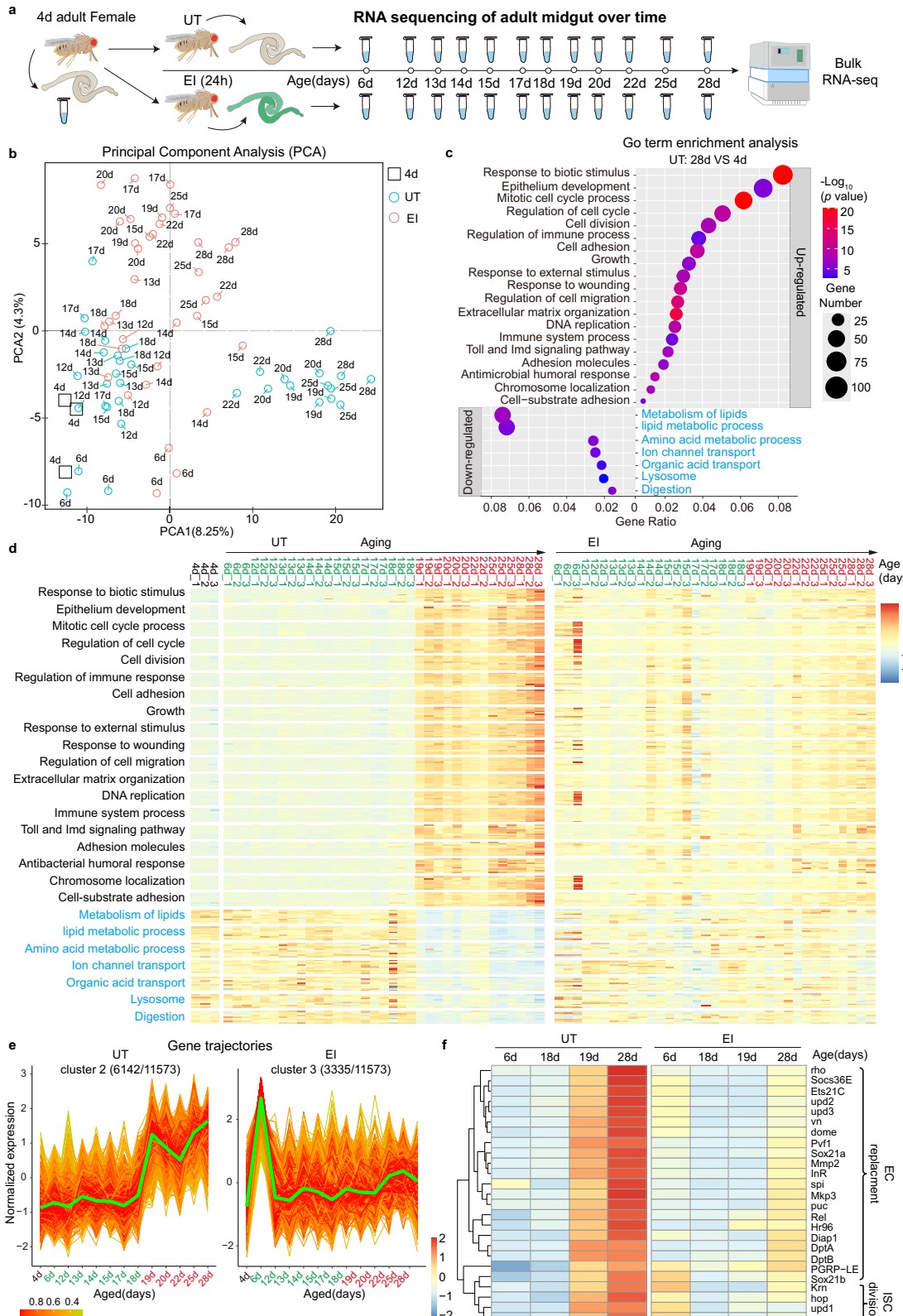

**Fig. 2 | EI alleviates abrupt transcriptome changes that occur at day 19.**
**a** Schematic illustration of mRNA sequencing of UT and EI (bleomycin-fed for 24 h at 4 d) midguts on a daily basis from 4 d to 28 d. 3 biological replicates for each day. **b** Principal component analysis (PCA) performed on the time course of the UT and EI transcriptomes. **c** Gene Ontology (GO) enrichment analysis for differentially expressed genes (p < 0.05, 2,719 genes) between the midgut of UT 4 d (young) and 28 d (old). *P* values was calculated by Fisher's exact test. **d** Expression heatmap analysis is performed chronologically from 4 d to 28 d for the top 10 genes of each

GO category in (**c**) of UT and EI transcriptomes. **e** Temporal trend gene trajectories of UT cluster 2 (6142 genes) show an abrupt upregulated change at 19 d, in contrast to gene trajectories of EI cluster 3 (3335 genes), which show flat changes after the EI. The green lines represent the average trajectory for each cluster. **f** The expression heatmap shows that genes involved in EC replacement and ISC division are abruptly up-regulated at 19 d in the UT midgut, while EI alleviates these gene up-regulations. Source data are provided with this paper.

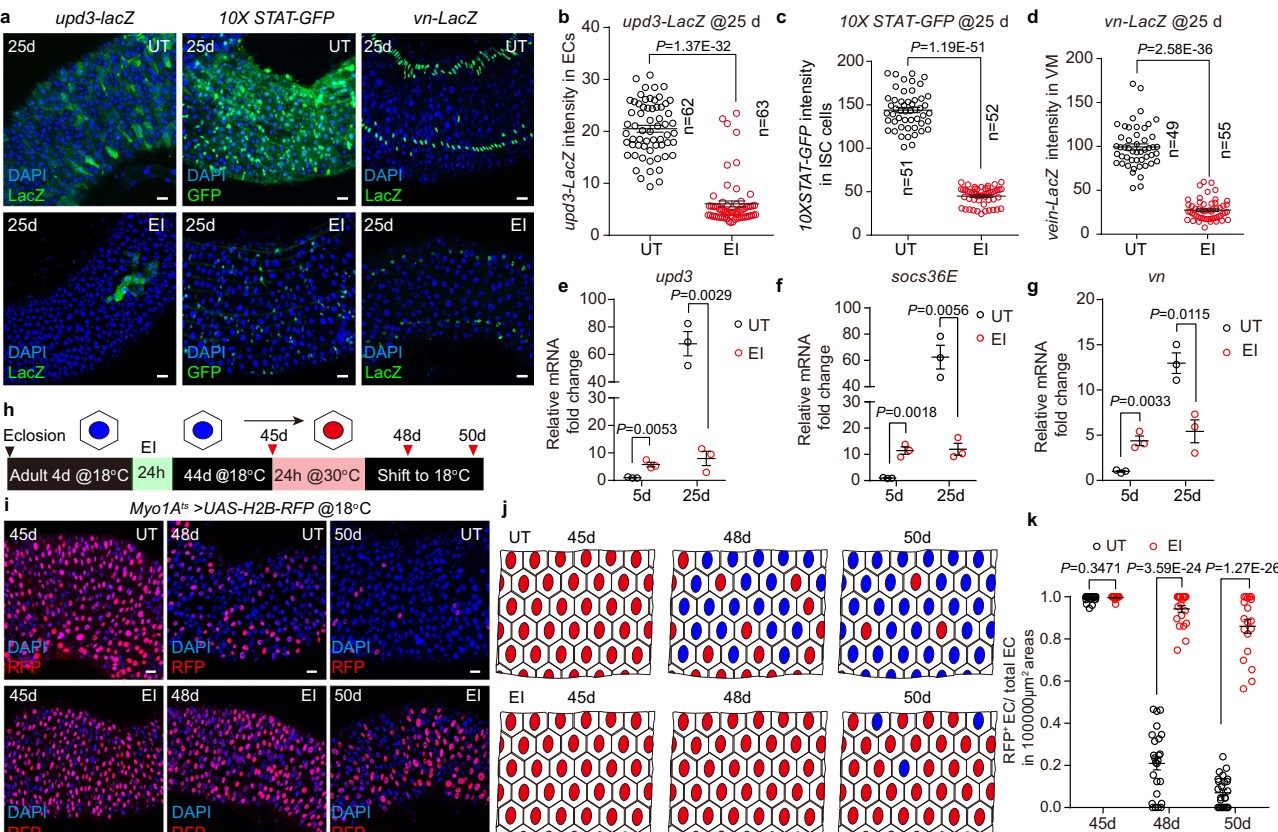

**Fig. 3 | EI reduces the EC turnover in aged flies.** Representative images (**a**) and intensity statistics (**b**–**d**) of *upd3-LacZ*, *10XSTAT-GFP*, and *vn-LacZ* in the midgut of 25 d UT and EI (bleomycin-fed for 24 h at 4 d) flies. n, number of scored cells. **e**–**g** *upd3*, *vn* and *socs36E* mRNA fold change in UT and EI midgut at 5 d and 25 d. n = 3 independent experiments. **h** Schematic illustration of the EC labeling and tracing in the aged midgut after the EI. EC labeling is achieved by transient expression of H2B-RFP in ECs using *Myo1A^{ts}* > *H2B-RFP* for 24 h. Representative images (**i**) and cartoon illustrations (**j**) of tracing of ECs by the transiently labeled H2B-RFP in UT and EI midguts at 45 d, 48 d and 50 d. Newly generated ECs are RFP⁻ and visualized by the large polyploid DAPI staining. **k** Statistics of the ratio of RFP⁺ ECs to total ECs in selected areas (10000 μm²) of UT and EI posterior midguts. n, numbers of regions. UT: n = 22 (45 d); 25 (48 d); 26 (50 d), EI: n = 21 (45 d); 22 (48 d); 19 (50 d). Data are mean ± SEM. Significance was determined using two-tailed unpaired *t* test. Scale bars, 20 μm. Source data are provided as a Source Data file.

In order to track EC turnover in the aged midgut, we transiently expressed the long half-life fluorescent protein H2B-RFP[30,77] in ECs by the temperature-controlled EC driver *Myo1A^{ts}* to label ECs and track their fate. UT and EI *Myo1A^{ts}* > *H2B-RFP* flies were cultured at 18 °C and transiently transferred to 30 °C for 24 hours at 44 d (an aged time point of UT flies) to activate H2B-RFP expression, after which the flies were reared at 18 °C to stop the expression of H2B-RFP (Fig. 3h). We found that most of the H2B-RFP-labeled (RFP⁺) ECs in the UT flies were replaced by the newly formed ECs lacking RFP labeling (RFP⁻) within 5 days (Fig. 3i–k). In the EI group, however, only a small fraction of the RFP⁺ ECs were replaced (Fig. 3i–k). Thus, our EC labeling and tracking experiments suggest that EI leads to a slowdown of EC turnover in the aged midgut.

## EI induced age mosaic prevents the abrupt onset of aging hyperplasia

So why does the UT midgut abruptly undergo a drastic EC turnover at about 20 days that leads to hyperplasia, while the EI midgut avoids these abrupt changes? Since young adult ECs were all generated simultaneously in the pupal stage[48,50,78], we tracked the first wave of EC turnover by labelling all ECs with H2B-RFP before eclosion (Fig. 4a). Consistent with previous findings that EC turnover is rarely observed in young healthy midgut[23,30,31], UT midgut reared on fresh food at 18 °C showed almost no EC turnover during the first 10 days, with a limited ratio of EC turnover

occurring in some midguts by day 14 (Fig. 4a, b, e and Supplementary Fig. 4a, b, e). By contrast, EI resulted in a loss of approximately 20% of puparium labeled RFP⁺ ECs by day 6, approximately 40% by day 10, and approximately 60% by day 14 (Fig. 4a, b, e and Supplementary Fig. 4a, b, e). Because all lost RFP⁺ ECs were replaced by RFP⁻ newly generated ECs, the midgut that underwent EI had its ECs in an age mosaic state that "old" ECs mixed with new ECs (Fig. 4b and Supplementary Fig. 4b).

To track the fate of ECs over a longer period of time, we again transiently transferred the fly to 30 °C at 22 d and labeled all ECs with H2B-RFP a second time (Fig. 4c and Supplementary Fig. 4c, d, f). The majority of UT ECs did not undergo turnover at 27 d and 37 d (Fig. 4c–e and Supplementary Fig. 4c, d, f). However, at 42 d almost all RFP⁺ ECs abruptly disappeared along with the onset of *esg*⁺/Arm⁺ cell hyperplasia (Fig. 4c–e and Supplementary Fig. 4c, d, f, g). By contrast, after the second labeling of the EI midgut, RPF⁺ ECs were gradually lost with fly age, with approximately 20% of RFP⁺ ECs lost at 27 d, 40% at 37 d and 50% at 42 d (Fig. 4c–e and Supplementary Fig. 4c, f), and the number of PH3⁺ cells at 42 d was significantly lower than that in the UT group (Fig. 4f and Supplementary Fig. 4c). Moreover, the *esg*⁺/Arm⁺ ISC-EB pairs in the 42 d EI midgut were in a homeostatic state (Supplementary Fig. 4c, d, f, g), suggesting that the hyperplasia was alleviated. Not only does EI cause EC in the midgut to form an age mosaic, but when we co-expressed H2B-RFP and *Reaper* by *Myo1A^{ts}* to label EC while inducing EC apoptosis, we similarly found an age mosaic of ECs (Fig. 4g, h).

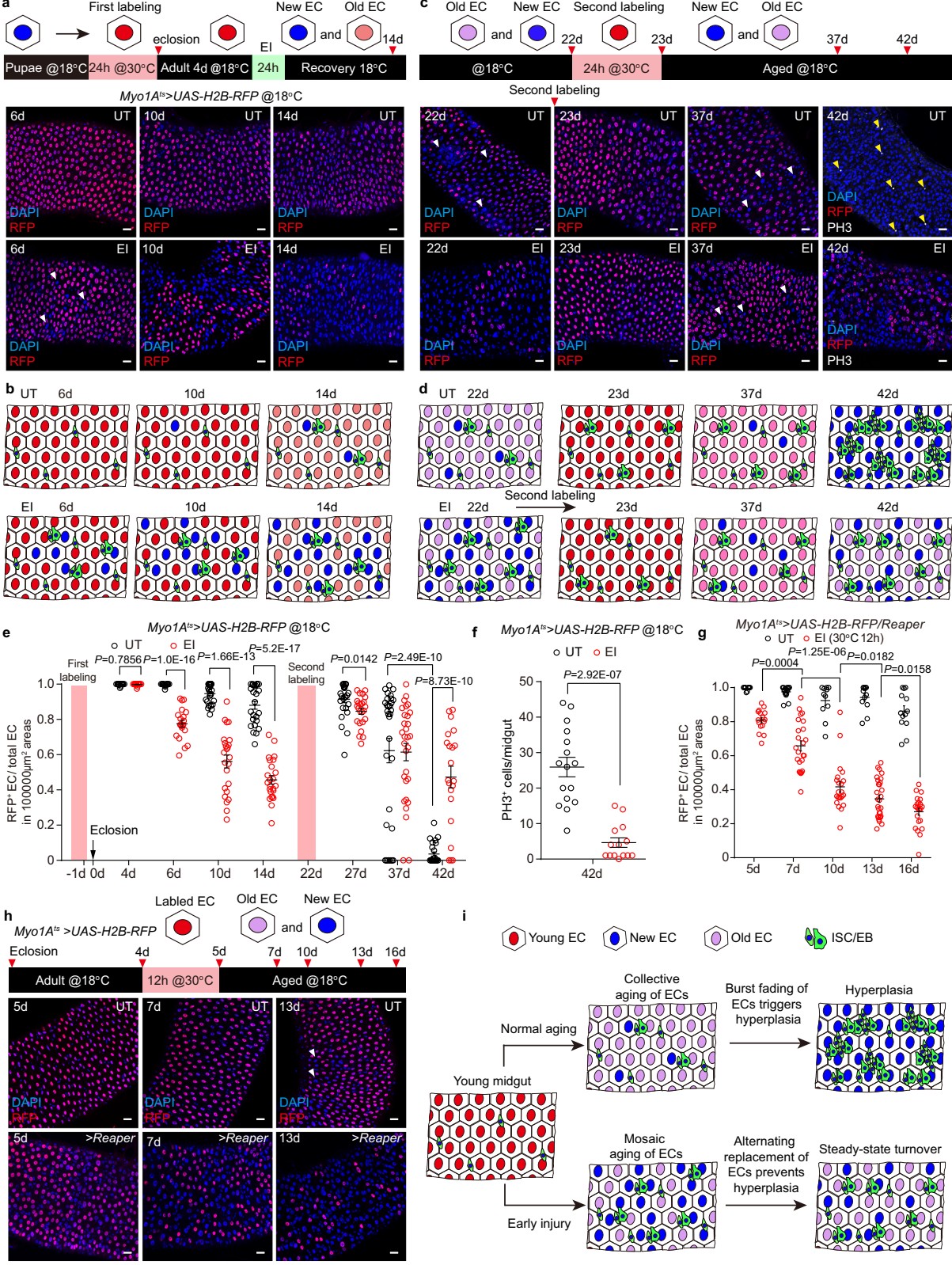

Taken together, our labeling and tracing experiments suggest that the majority of puparium-generated ECs in the UT midgut undergo collective turnover at a given aging time point, triggering midgut hyperplasia, whereas EI avoids collective EC death by introducing new ECs during the aging process to form an aging mosaic in an alternating replacement manner, preventing the onset of hyperplasia (Fig. 4i).

**ROS rise and Lamin loss lead to collective EC death**

To understand why the puparium-generated ECs collectively turnover, we compared the gene expression difference between 18 d and 4 d transcriptomes of the UT group that preceded the abrupt hyperplasia. Go term enrichment analysis of 18 d upregulated genes revealed enrichment of transcripts involved in microbial defense, immune response and reactive oxygen species (ROS) metabolic (Fig. 5a). A daily

**Fig. 4 | The age mosaic induced by EI prevents the abrupt onset of aging hyperplasia. a** Upper, schematic illustration of the method to track the turnover of puparium labeled H2B-RFP[+] ECs in UT and EI (bleomycin-fed 24 h at 4 d) midguts. Lower, representative images of the tracing pattern of H2B-RFP[+] ECs in UT and EI midguts at 6 d, 10 d and 14 d. Midgut regions were selected from R4C/R5A. Newly generated ECs (white arrowheads) are RFP[+] and visualized by the large polyploid DAPI staining. **b** Cartoon showing the distribution of puparium-labeled RFP[+] (red) ECs versus newly generated RFP[-] (blue) ECs in the UT and EI midgut during aging. The red color, representing the amount of H2B-RFP in the EC nuclear, fades with age. Green cells are ISC-EB pairs. **c** Upper, schematic illustration of the second H2B-RFP labeling and tracing of ECs. Lower, representative images of the tracing pattern of RFP[+] ECs in UT and EI midguts after 22 d. White arrowheads indicate the newly generated RFP[-] ECs, yellow arrowheads indicate PH3[+] cells. **d** Cartoon showing the abrupt collective turnover of RFP[+] ECs from 37 d to 42 d, contrasting with the formation of the age mosaic of EI midgut. **e** Statistics of the ratio of RFP[+] ECs to total ECs in (**a**) and (**c**). The red stripe indicates the duration of EC labeling. n, numbers of regions, UT: n = 24 (4 d), 26 (6 d), 25 (10 d), 25 (14 d), 26 (27 d), 30 (37 d), 26 (42 d); EI: n = 22 (4 d), 21 (6 d), 24 (10 d), 24 (14 d), 24 (27 d), 30 (37 d), 21 (42 d). **f** Statistics of the number of PH3[+] mitotic cells in UT and EI midgut at 42 d. n, numbers of midguts. n = 16 (UT), 14 (EI). Statistics of the ratio of RFP[+] ECs to total ECs (**g**) and representative images of the tracing pattern of H2B-RFP[+] ECs in UT and *Myo1A[ts]>Reaper* midguts (**h**). n, numbers of regions. UT: n = 10 (5 d), 17 (7 d), 11 (10 d), 12 (13 d), 13 (16 d); EI: n = 16 (5 d), 24 (7 d), 22 (10 d), 29 (13 d), 22 (16 d). White arrowheads indicate occasional newly formed ECs in the UT midgut. **i** A model illustrates how EI induced EC age mosaic prevents the abrupt onset of aging hyperplasia. Data are mean ± SEM. Significance was determined using two-tailed unpaired *t* test. Scale bars, 20 μm. Source data are provided as a Source Data file.

basis gene expression heat map shows that these biological processes have an increasing trend with age from 4 d to 18 d (Fig. 5b), supporting the increasing evidence that gut microbes induce immune responses and ROS accumulation in the gut epithelium[19,21,23,79–86]. Since infection-induced ROS leads to EC turnover[20,23,33,87], we speculate that the continuous accumulation of ROS in puparium-generated ECs results in collective EC death.

To directly observe ROS accumulation in puparium-generated ECs, we monitored *gstD1-GFP*, which is used to indicate the level of ROS activation[88], in puparium labeled RFP[+] ECs (Fig. 5c). We found that as flies aged, *gstD1-GFP* accumulated in RFP[+] ECs of UT midgut (Fig. 5d and Supplementary Fig. 5a, b). By contrast, in the EI 30 d midgut, the GFP signal was significantly reduced in the newly generated RFP[-] ECs (Fig. 5d, e and Supplementary Fig. 5b), forming a mosaic of ROS levels. Similarly, the ROS fluorescent probe H2DCF[89] also showed that ROS levels in the RFP[+] cells accumulated with age, and there was a lack of H2DCF staining in EI introduced young RFP[-] ECs (Supplementary Fig. 5c, d). To demonstrate that ROS accumulation causes EC turnover and gut hyperplasia, we upregulated ROS levels by knockdown a negative redox regulator CncC[23,88] in young ECs. Knockdown of *CncC* resulted in dramatic upregulation of redox-sensitive dye dihydro-ethidium (DHE)[23,89,90] staining and EC turnover marker *upd3-LacZ* staining in young ECs (Supplementary Fig. 5e, f), H2B-RFP labeled EC turnover (Supplementary Fig. 5g, h) and the onset of hyperplasia (Fig. 5f). Conversely, removal of the intracellular ROS by feeding flies with the ROS scavenger N-Acetyl Cysteine (NAC)[89] alleviated the hyperplasia in aged midgut (Supplementary Fig. 5i, j), suggesting that blocking the accumulation of ROS prevents collective EC turnover. Taken together, our data suggest that puparium-generated ECs accumulate ROS upon gut microbial exposure and reach a threshold at approximately 20 d (25 °C), triggering collective turnover. By contrast, EI-induced new ECs are in a low ROS state, preventing collective EC turnover.

It is reported that ROS production inhibits Lamin expression[91], and Lamin loss is an age-related marker in response to multiple cellular stresses[92–95]. Therefore, we examined changes in Lamin expression over time in puparium-generated ECs. In the 4 d UT midgut, strong Lamin staining was observed on the nuclear membranes of both diploid progenitors and puparium H2B-RFP labeled polyploid ECs (Fig. 5g and Supplementary Fig. 6a–c)[96,97]. Notably, Lamin staining on the nuclear membranes of RFP[+] ECs gradually disappeared with age (Fig. 5g, h) and was absent in RFP[+] ECs in the 30 d epithelium (Fig. 5g and Supplementary Fig. 6a). However, in the EI 30 d midgut, Lamin staining was restored in newly generated RFP[-] ECs, suggesting that Lamin staining on the nuclear membrane is a marker of young ECs (Fig. 5g, i). Furthermore, knockdown of Lamin in young ECs results in midgut hyperplasia (Fig. 5j and Supplementary Fig. 6d), indicating that the loss of Lamin in the puparium-generated ECs could be the cause of the collective EC turnover.

We also examined the expression of the A-type Lamin protein, Lamin C (LamC)[94,98,99], during the aging process. However, unlike the expression of Lamin, LamC was mainly expressed in ECs and showed no age difference (Supplementary Fig. 6g, h). Moreover, knockdown of LamC in ECs did not cause midgut hyperplasia (Supplementary Fig. 6e, f), suggesting that Lamin, but not LamC, plays a key role in EC aging and turnover.

Taken together, we propose that after eclosion, puparium-generated ECs possess high levels of Lamin to maintain their healthy physiological functions. Subsequently, ROS accumulated in response to gut microbes and Lamin gradually degenerated, triggering a collective turnover when the threshold was reached; nevertheless, EI-induced new ECs restored the original Lamin levels and low ROS status, avoiding collective death with old ECs (Fig. 5k).

## Age mosaic prevents the disruption of the septate junction caused by the collective EC death

We are curious as to why ISC hyperplasia persists once EC collective turnover occurs, but the age mosaic of ECs continues to prevent hyperplasia. For the maintenance of the intestinal barrier and homeostasis, an intact septate junction (SJ) is essential[26,27,29]. Consistent with previous findings that SJ progressively decreases with fly aging[26], when ECs were labeled as RFP[+] in the puparium (Fig. 6a), we found that staining of SJ proteins Discs large 1 (Dlg1) and Coracle (Cora) among RFP[+] ECs in the UT midgut was significantly reduced at 18 °C rearing for 30 days (Fig. 6b, c and Supplementary Fig. 7a–d). By contrast, Dlg1 and Cora staining of RFP[+] ECs in the 30 d EI-induced age mosaic epithelium was significantly increased (Fig. 6b, c and Supplementary Fig. 7a–d). We also examined the adhesion junction protein E-cadherin and found that it did not change with age or in the age mosaic (Supplementary Fig. 7e, f). Furthermore, at 37 d, after the UT midgut had undergone hyperplasia (Fig. 6d), we found that Dlg1 staining represented SJs that were disrupted and disorganized in the UT midgut, whereas SJs were intact and regularly arranged in the EI midgut with age mosaic (Fig. 6e, f), suggesting that age mosaic improves SJs between ECs.

Recent studies have shown that when an EB differentiates into a new EC, the EB first contacts the basal terminus of SJs between mature ECs, thereafter SJs extend to cover the apical membrane of this pre-EC, thus forming new SJs that circumscribe the apical surface between the new EC and the surrounding mature ECs[42,43]. We therefore speculate that EI-induced new RFP[-] ECs reconstitute SJs with old RFP[+] ECs. Our results showed that the level of Dlg1 staining of the RFP[+] boundary old (b old) ECs was significantly higher when they were in contact with RFP[-] new ECs than RFP[+] ECs (old ECs) that were only in contact with old ECs (Fig. 6g, h), indicating that the new ECs improved the SJs of the old ECs surrounding them.

To demonstrate that EI-induced age mosaic prevents midgut hyperplasia by improving SJs, we knocked down Dlg1 and three other SJ components (Cora, Mesh and Tsp2A)[29] in EC after EI induced ISC division was restored to the quiescent state, respectively (Fig. 6i and

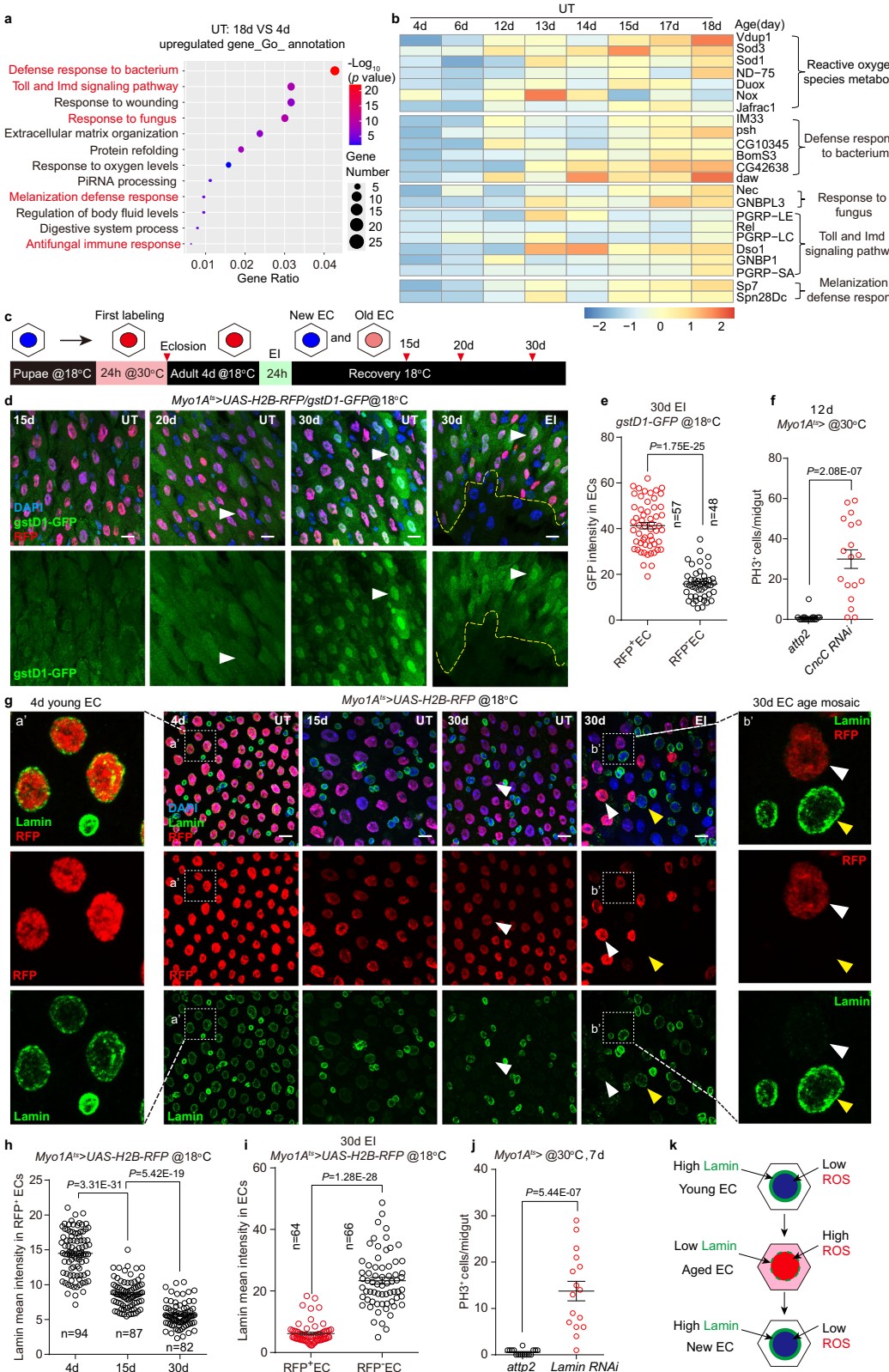

Supplementary Fig. 7g–i). Deficiency of SJs all resulted in hyperplasia in the EI midgut (Fig. 6i and Supplementary Fig. 7g–i), suggesting that the integrity of SJs is required for midgut homeostasis with age mosaic.

It is conceivable that an EB cannot form intact SJs with ECs undergoing continuous collective turnover, and that EC turnover stimulates ISCs to proliferate and continuously generate new EBs, resulting in hyperplasia formation. If this is the case, then suspending the division of ISCs may allow the newly generated ECs to have the time and space to repair the disrupted SJs (Fig. 6j). Therefore, we temporarily halted (TH) ISC mitosis of aged hyperplastic midguts (65 d at 18 °C) by transferring $esg^{ts}$>$Cdk2$ $RNAi$ flies to 30 °C for 7 days[100]. The effect of $Cdk2$ RNAi on mitosis was then removed by transferring the flies to 18 °C (Fig. 6k). We found that SJs in midgut subjected to TH treatment acquired significant improvement at 85 d (Fig. 6k, l and

**Fig. 5 | Increase in ROS and loss of Lamin leads to collective puparium-generated EC death. a** Differentially expressed genes (p < 0.05, 1094 genes) between 4 d and 18 d UT midgut are classified based on GO terms for biological process and KEGG pathway. The microbiome defense and immune response categories are shown in red. *P* values was calculated by Fisher's exact test. **b** Expression heatmap analysis is performed chronologically from 4 d to 18 d for the ROS metabolic and microbiome defense pathways of UT transcriptomes. **c** The schematic shows the method for monitoring ROS levels in puparium-generated ECs. ECs are transiently labeled with H2B-RFP prior to eclosion. The newly generated ECs in the adult are RFP- polyploid cells (blue nuclear ECs). **d** Representative images of *gstD1-GFP* staining in puparium-generated ECs (RFP+, white arrowhead) and newly generated ECs (RFP-) in UT and EI midgut during aging. Note in the EI 30 d midgut, the boundary between *gstD1-GFP* intensity is the boundary between RFP+ and RFP- ECs (the yellow dash line). **e** Statistics of *gstD1-GFP* intensity in RFP+

ECs or RFP- ECs in 30 d EI midgut. n, numbers of ECs. **f** Statistics of the number of PH3+ cells in *Myo1A*ts driving *attp2* (control) and *CncC RNAi* midguts. n, number of midguts. n = 19 (*attp2*), 18 (*CncC RNAi*) midguts. **g** Representative images of Lamin staining in puparium-generated ECs (RFP+, white arrowhead) and newly generated ECs (RFP-, yellow arrowhead) in the UT and EI midgut during aging. Close-up inset **a'** shows strong Lamin staining in 4 d young RFP+ ECs, inset **b'** shows Lamin staining is absent in RFP+ ECs but is restored in RFP- ECs in 30 d EI midgut. **h** Statistics of Lamin intensity in RFP+ ECs during aging. n, numbers of ECs. **i** Statistics of Lamin intensity in RFP+ or RFP- ECs in 30 d EI midguts. n, numbers of ECs. **j** Statistics of the number of PH3+ cells in *Myo1A*ts driving *attp2* (control) and *Lamin RNAi* midguts. n = 16 (*attp2*), 16 (*Lamin RNAi*) midguts. **k** A schematic summary of ROS and Lamin dynamics during EC aging and replacement. Data are mean ± SEM. Significance was determined using two-tailed unpaired *t* test. Scale bars, 10 μm. Source data are provided as a Source Data file.

Supplementary Fig. 7j, k). Correspondingly, hyperplasia was alleviated in TH-treated midgut up to 95 d (Fig. 6m, n). Since ISCs in 87 d TH-treated midgut still overproliferated after *Ecc15* feeding (Supplementary Fig. 7l), excluding the possibility that *Cdk2 RNAi* expression leads to ISC senescence, our TH experiments demonstrate that restoration of SJs in aged midgut can reverse the progression of hyperplasia. Furthermore, we found that TH treatment of aged midgut (65d-72d) significantly improved the barrier function of 100 d and 115 d midgut (Fig. 6o) and significantly prolonged the lifespan of aged flies (Fig. 6p), achieving a reversal of advanced aging.

In summary, our results suggest that age mosaic prevents hyperplasia in the aged midgut by continuously repairing the SJ through alternate EC replacement. Consistently, forcing the formation of intact SJs in the aged midgut could prevent hyperplasia and reverse the aging process.

## iTRF-induced age mosaic leads to alleviated hyperplasia and extended lifespan

We wondered how we could create an age mosaic in the gut epithelium without damaging the gut. Recent studies have shown that compared to ad libitum (Ad) feeding, intermittent time-restricted feeding (iTRF) prevents midgut hyperplasia in aged flies, enhances the integrity of the midgut barrier, and robustly extends the lifespan of flies[101,102]. The similarity of these physiological effects between iTRF and EI treatment led us to wonder if iTRF similarly contributes to the creation of an age mosaic in the midgut epithelium. We first tested the iTRF regimen[101], which consisted of fasting the flies from Zeitgeber time 6 for 20 h every other day from 10 d to 30 d, and resuming the Ad diet for 28 h between fasting days and after 30 d (Fig. 7a). We confirmed that iTRF improved midgut barrier function in aged flies (Supplementary Fig. 8a) and significantly extended lifespan (Supplementary Fig. 8b). Then we examined the number of PH3+ dividing ISCs in Ad and iTRF midguts in a time course. Remarkably, we found that the number of PH3+ cells was significantly increased during the iTRF fasting phase, and returned to a low level on a Ad diet in the aged flies (Fig. 7b). Consistent with the increase of PH3+ cells, the puparium labeled H2B-RFP+ ECs were continuously replaced by newly generated RFP- ECs during the iTRF fasting phase, creating an age mosaic in the midgut epithelium (Fig. 7c, d). Notably, day-biased iTRF fast did not result in EC turnover (Supplementary Fig. 8c–e), suggesting that the formation of EC age mosaic requires night-biased fasting.

Previous report has shown that the health benefit of iTRF requires night-biased circadian autophagy[101]. To test whether the autophagy is sufficient to induce EC turnover, we overexpressed the autophagy component *Atg1* in Ad fed ECs during the night and found that *upd3* expression was dramatically activated in ECs, accompanied with a significant increase in ISC proliferation and generation of the EC age mosaic (Fig. 7e and Supplementary Fig. 9a–d). By contrast, feeding flies with the autophagy inhibitor chloroquine (CQ) during the fasting phase of iTRF blocked the generation of age mosaic (Supplementary

Fig. 9e–g). The above results indicated that iTRF induced autophagy leads to EC turnover and formation of an age mosaic. Furthermore, to demonstrate that the alleviation of hyperplasia by iTRF requires autophagy in ECs, we knocked down *Atg1* and *Atg8a*, respectively, during iTRF fasting and found that inhibition of autophagy in ECs prevented ISC proliferation during the fasting period (Fig. 7f, g). Correspondingly, while hyperplasia did not occur in the 70 d control iTRF midgut, hyperplasia occurred in the 70 d midgut in which autophagy was blocked during the fasting period (Fig. 7f, h), demonstrating that autophagy-generated age mosaic is required for iTRF to alleviate midgut hyperplasia in aged flies.

Our results showed that the iTRF-induced EC age mosaic significantly improved Dlg1 staining among ECs in the aged midgut (Fig. 7i, j and Supplementary Fig. 9h, i). To test whether the iTRF-induced health benefits depend on the age mosaic, we inhibited ISC division during the iTRF fasting period (Fig. 7k) and found that blocking the age mosaic formation resulted in iTRF failing to improve septate junctions (Supplementary Fig. 9j–l) and failing to alleviate hyperplasia in the aged midgut (Fig. 7k). Meanwhile, the effect of iTRF on lifespan extension was also significantly reduced (Fig. 7l). Taken together, our data suggest that the health benefits of iTRF require night-biased autophagy to induce EC aging mosaic, which improves SJ and alleviates midgut hyperplasia.

## Discussion

Inflammatory responses and hyperplasia that occur in the aged *Drosophila* midgut as markers of aging have been intensively studied[5,6,8,17,21]. However, due to discontinuities in the selection of studied time points[8,21,62,65,103] and the finding of a linear increase in gut microbial defense in early adult stages[21,23,83,84], aging of the *Drosophila* midgut is considered by default to be a linear process. Here, by continuous daily dissection, we unexpectedly found that the onset of midgut hyperplasia is an abrupt event. Daily transcriptome sequencing suggests that nonlinear hyperplasia results from massive turnover of EC cells in a single day. We found that under room temperature rearing conditions in the laboratory, ECs generated simultaneously during the pupal period undergo a collective turnover at around 20 days, triggering the onset of hyperplasia. Thus, the fact that ECs have a fixed lifespan (under certain physiological conditions) determines the nonlinearity of the aging process.

Recent studies have revealed nonlinear dynamic changes in transcriptome and proteomics during human aging[104–107]. We are curious whether there is also a fixed lifespan for human cells, which would result in a sudden acceleration of aging when key cell populations reach the end of their lives. Our study may inspire an understanding of the nonlinear mechanisms of human aging at the age level of tissue cells.

In our study, collectively dying ECs at day 20 release high levels of the JAK-STAT signaling ligands Upd2 and Upd3, resulting in the onset of hyperplasia (Fig. 3). According to Cai et. al, ligands released

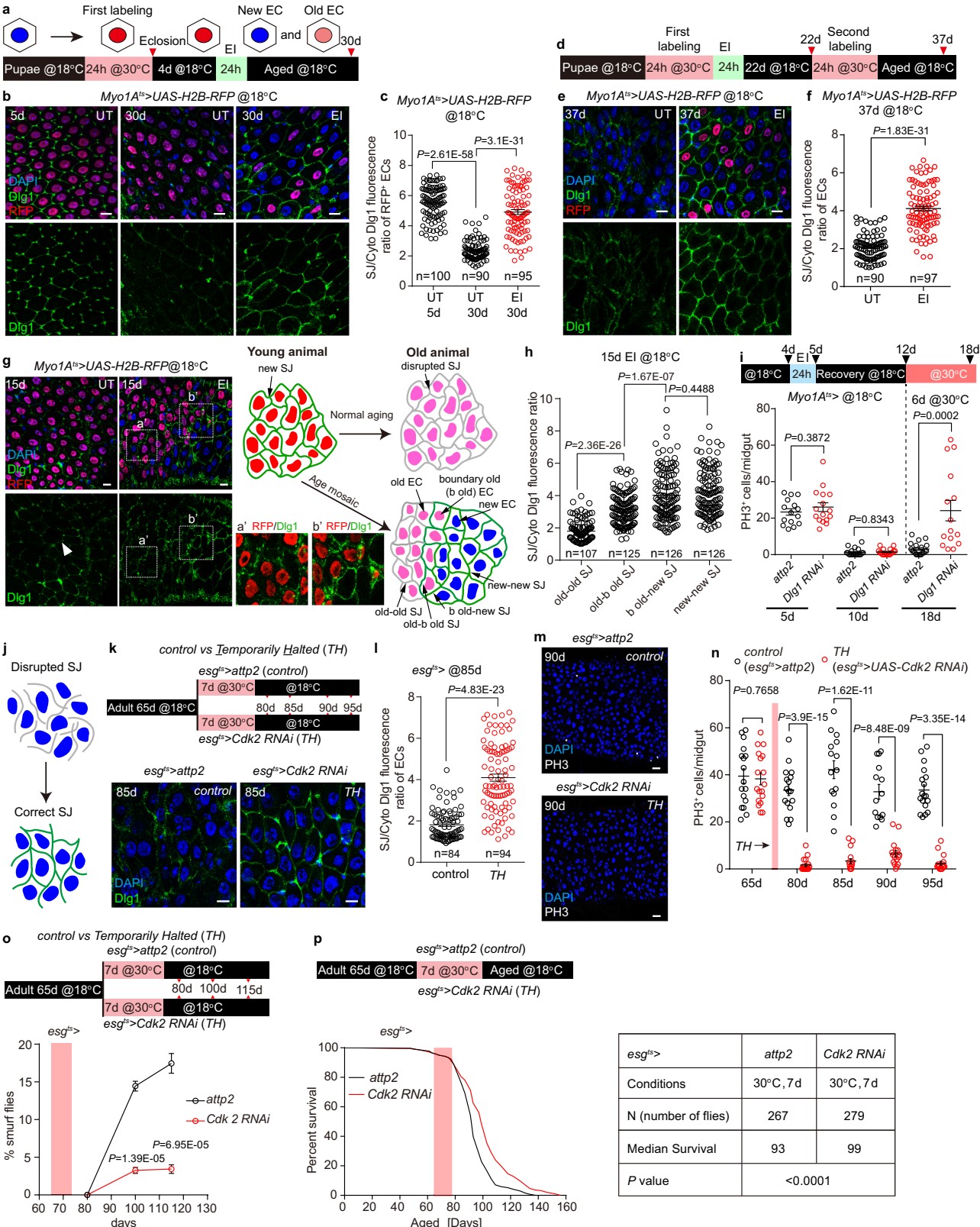

by ECs activate JAK-STAT signaling in ensheathing glial cells, inducing reprogramming of lipid metabolism in antennal lobe glial cells and neurons, causing a decrease in olfactory discrimination[108]. Notably, senescent glial cells begin to appear in the antennal lobes at around day 20, and these emerging senescent glial cells alter lipid metabolism[109]. Thus, the collective turnover of ECs at day 20 provides an explanation for the timing of the emergence of senescent

glial cells in the antennal lobes, suggesting that brain aging originates in the gut epithelium.

Early injury induced EC age mosaic improves gut barrier function in aged flies and extends lifespan, echoing German philosopher Friedrich Nietzsche's aphorism that "what does not kill us makes us stronger." It has long been known that exposure to low doses of noxious chemical or physical agents during the normal aging process

**Fig. 6 | Age mosaic improves the septate junctions in aged midgut. a** The schematic shows the method for monitoring SJs in the midgut with puparium H2B-RFP labeled ECs. Representative images (**b**) and statistics (**c**) of the Dlg1 staining in UT and EI (bleomycin-fed 24 h at 4 d) midgut during aging. **d** Schematic illustration the second EC labeling and tracing of ECs in UT and EI midgut. Representative images (**e**) and statistics (**f**) of the Dlg1 staining in UT and EI midgut at 37 d. **g** Representative images of the Dlg1 staining in 15 d UT and EI midgut. Three types of ECs are present in the EI midgut, RFP⁻ new ECs, RFP⁺ boundary old ECs adjacent to new ECs (b old), and RFP⁺ old ECs not adjacent to new ECs (old). Close-up views show Dlg1 staining in the region of old-old EC contacted (a′) and b old-new EC contacted (b′). Cartoon illustrates the observation that new ECs (RFP) restore SJs (green) in b old ECs (RFP⁺). **h** Statistics on the intensity of Dlg1 staining in four SJs generated between three different ECs in (**g**). **i** Upper, schematic of the method that knockdown *Dlg1* after EI (bleomycin-fed 24 h at 4 d). Lower, statistics of the number of PH3⁺ cells in *attp2* (control) and *Dlg1* knockdown midgut at 5 d, 10 d and 18 d. n, number of midguts. *attp2*: n = 15 (5 d), n = 19 (10 d), n = 20 (18 d), *Dlg1* knockdown: n = 16 (5 d), n = 18 (10 d), n = 15 (18 d). **j** A cartoon shows the hypothesis that halting the hyperplasia restores the correct SJs among ECs. **k** Upper, schematic of temporarily halting (TH) the ISC division in 65 d hyperplastic midgut by *esgᵗˢ>Cdk2 RNAi* for 7 days. Lower, representative images of Dlg1 staining in control and TH midgut at 85 d. **l** Statistics of Dlg1 staining in 85 d control and TH midgut. Representative images (**m**) and statistics (**n**) of PH3⁺ cells in aged control and TH midgut. n, number of midguts. control: n = 17 (65 d), 17 (80 d), 15 (85 d), 14 (90 d), 18 (95 d); TH: n = 18 (65 d), 18 (80 d), 17 (85 d), 16 (90 d), 17 (95 d). **o** Upper, schematic of a short-term genetic manipulation that inhibits the excessive proliferation of intestinal stem cells in aged flies. Lower, statistics of percentage of smurf flies in the control (*esgᵗˢ>attp2*) and TH files group. The red stripe indicates the duration of TH. n, number of flies. Control: n = 84, 97, 80. TH: n = 74, 93, 104. **p** Survival analysis of flies in control (*esgᵗˢ>attp2*) and TH group. The red stripe indicates the duration of TH. Data are mean ± SEM. Significance was determined using two-tailed unpaired *t* test. n, number of measured ECs (**c**, **f**, **l**), number of SJs (**h**), or number of midgut (**i**, **n**). Scale bars, 10 μm (**a**, **e**, **g**, **f**, **k**) or 20 μm (**m**). Source data are provided as a Source Data file.

induces hormesis effects that confer stress resistance and extend lifespan[110]. For example, transient exposure to low concentrations of oxidants during development extends adult lifespan in *Drosophila*[111] or *C. elegans*[112]. In mice, a prolonged memory to acute inflammation enables mouse EpSCs to accelerate barrier repair after subsequent tissue damage in mouse skin[113]. Dietary restriction (DR), short-term intermittent fasting (IF) and intermittent time-restricted feeding (iTRF) could be viewed as stressful conditions leading to hormesis[110]. Recent studies have shown that DR, IF and iTRF achieve health benefits at least in part by ameliorating age-related gut dysfunction[101–103,114–119]. Although studies suggest that fasting reduces age-related intestinal pathologies by suppressing mTOR signaling and increasing autophagy in the gut epithelium[101,103,119], it is still unclear how activation of autophagy in epithelial cells attenuates hyperplasia and improves barrier function. Our results show that iTRF-induced activation of autophagy is sufficient to induce EC turnover in the *Drosophila* midgut, thereby generating an EC age mosaic in the midgut epithelium, preventing collective EC turnover and establishing alternative EC replacement to improve gut barrier function. Thus, our age mosaic model provides a detailed mechanism for how fasting improves the health of the aged gut.

Although the mouse small intestine, like the *Drosophila* midgut, undergoes age-related loss of microbial homeostasis, increased intestinal permeability and systemic inflammation with aging[11], the current studies suggest that the newly generated ECs migrate upward and shed into the lumen when they reach the tip of the villi[120–122], without considering whether there is an age mosaic of ECs[123]. However, the uneven generation of new ECs by TA cells surrounding the villi and the active EC migration may also create an age mosaic[122], in which the integrity of the intestinal epithelium may be affected. Indeed, using ¹⁵N isotope-tracing method to measure the age of cells and proteins in mouse organs, Martin Hetzer's lab showed that most organs are mosaics of cells of different ages[124]. Even the liver, which has a high turnover rate, contains cells that are as old as the animal[124]. Our results in *Drosophila* midgut suggest that age mosaicism may be a way for mammalian tissues to maintain an orderly tissue renewal against aging, and generating age mosaic may become a therapeutic approach to reverse aging.

## Methods
### Drosophila husbandry
Standard cornmeal food was prepared according to the following recipe: 210 g dry inactivated yeast, 900 g yellow cornmeal, 120 g soy flour, 100 g agar (Solarbio, cat# A8190), 800 ml light corn syrup, 0.3 g Levofloxacin (Dalian Meilun), 15 g Methyl 4-Hydroxybenzoate (Biosharp) and 12 L water. Flies were maintained at 25 °C and 65% humidity on a 12 h light/dark cycle, food changed every 2 days. Unless otherwise indicated, only female flies were used in all our experiments. 15-20 female flies were co-cultured with 5-8 young males in vials and 50-60 female flies were co-cultured with 15-20 young males in bottles[55,56].

### Drosophila stocks
The following stocks were used: *w¹¹¹⁸* (VDRC#6000), *esg-Gal4 UAS-GFP/CyO*, *esgᵗˢ: esg-Gal4 UAS-GFP tub-Gal80/CyO* (a kind gift from Bruce Edgar, The Utah University)[32], *tub-Gal80ᵗˢ* (a kind gift from Benjamin Ohlstein lab, Columbia University Medical Center)[14], *esg-GFP* (a kind gift from Benjamin Ohlstein lab, Columbia University Medical Center), *Gbe + Su(H)-LacZ (NRE-lacZ)* (a kind gift from Benjamin Ohlstein lab, Columbia University Medical Center)[125], *Myo1Aᵗˢ: Myo1A-Gal4 UAS-GFP tub-Gal⁸⁰/CyO* (a kind gift from Bruce Edgar, the university of Utah)[32], *Myo1A-Gal4 UAS-GFP tub-Gal⁸⁰/CyO UAS-H2B-RFP/TM6B* (this paper), *esg-Gal4 UAS-GFP/Cyo; UAS-H2B-RFP tub-Gal80ᵗˢ/TM6B* (a kind gift from Maria Dominguez lab at Consejo Superior de Investigaciones Científicas (CSIC))[77], *Mex-Gal4* (Obtained from HY. Chen lab at Sichuan University, BL#91367), *Mex-Gal4/CyO; UAS-H2B-RFP/tublin-Gal80ᵗˢ* (this paper), *UAS-Reaper* (Bloomington Drosophila Stock Center BL#5824) and *UAS-Atg1* (a kind gift from Zongzhao Zhai, Hunan Normal University). *UAS-Hepᵃᶜᵗ* (BL#9306) and *UAS-Hopᵗᵘᵐˡ* (a kind gift from Benjamin Ohlstein lab, Columbia University Medical Center), *Notchˢˢᵉˡˡ/FM7* (a kind gift from Gary Struhl lab, Columbia University Medical Center), *vn-LacZ/TM3* (BL#11749), *attP2* (BL#36303), *10xStat-GFP/TM6B* (BL#26198), *upd3-LacZ* (a kind gift from Bruce Edgar, The Utah University)[32], *UAS-Lamin RNAi* (a kind gift from Haiyang Chen, Sichuan University, VDRC#45635)[94], *E-cadherin-GFP* (a kind gift from HP. Yu lab at Southern University of Science and Technology)[126], *UAS-Cdk2 RNAi* (THU#201500107.s)[53], *UAS-Dlg1 RNAi* (Tsinghua Fly Center,THU#1938), *UAS-Mesh RANi* (NIG-Fly, 12074R-1)[29], *UAS-Tsp2A RNAi* (NIG-Fly, 11415R-2)[29], *gstD1-GFP/CyO* (a kind gift from Wanzhong Ge, Zhejiang University)[88], *UAS-Atg1 RNAi* (BDSC, BL44034)[101], *UAS-Atg8a RNAi* (BL#34340)[101], *UAS-LamC RNAi* (Tsinghua Fly Center, THU#201500538.S).

### Bleomycin, paraquat treatment and *Ecc15* infection
We adapted the midgut injury assay to the method described[20,47,57]. Filter paper was soaked with a solution consisting of 25 μg/mL bleomycin (MedChemExpress, HY-17565A) or 10 mM paraquat (Sigma-Aldrich, 36541) dissolved in 5% sucrose, untreated (UT) flies were fed with 5% sucrose. Bacterial strains *Erwinia carotovora carotovora15* (*Ecc15*) were cultured overnight in LB medium containing antibiotic (100 μg/mL rifampicin) at 30 °C with shaking overnight, and harvested by centrifugation at 3000 g for 30 minutes. the pellet was then re-suspended in the 1 mL 5% sucrose (at final OD₆₀₀100-200). Before treatment, flies were starved in empty vials for 2 h and then transferred

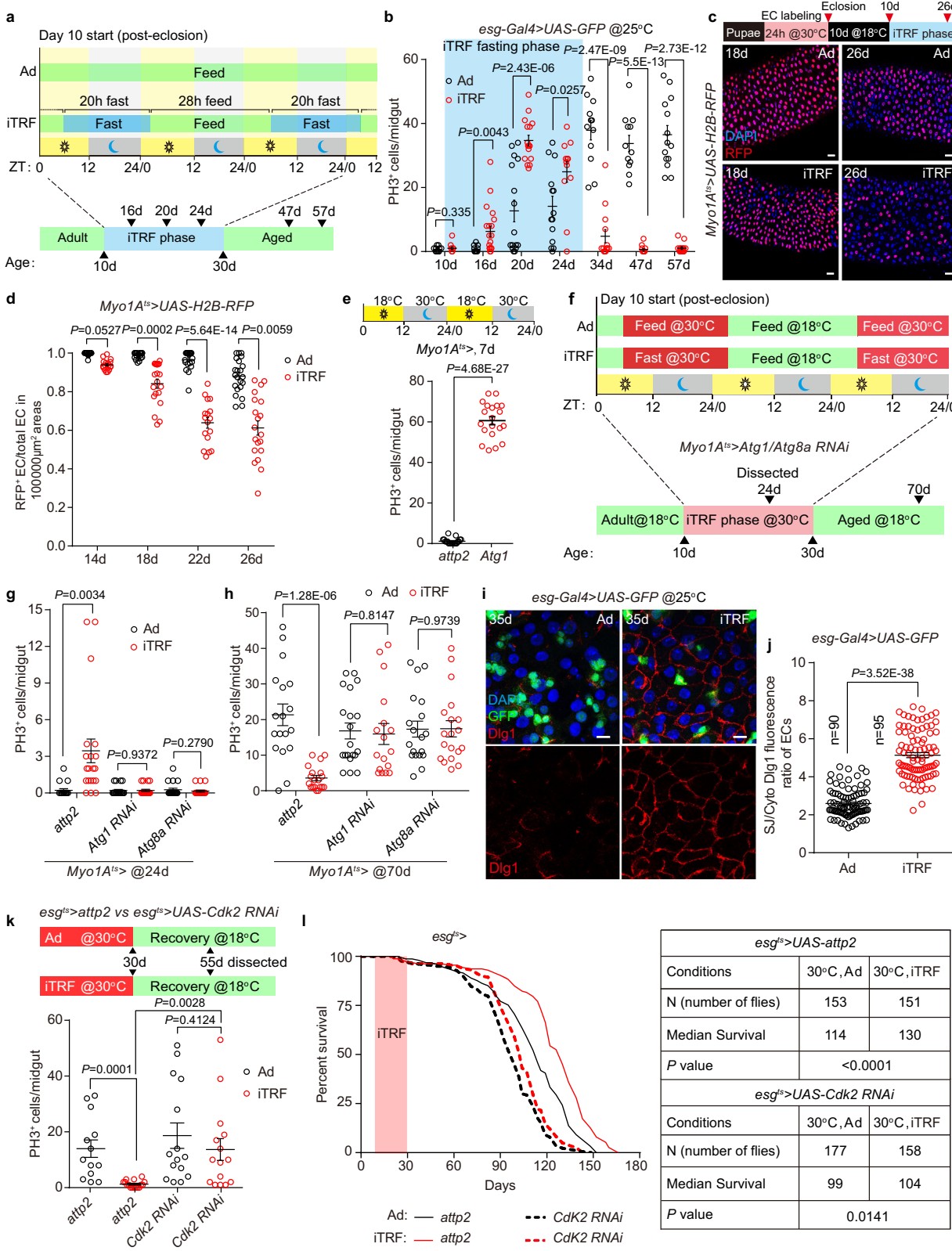

into vials with bleomycin, paraquat, or *Ecc15* solution–saturated paper. After 24 h of treatment, flies were transferred to fresh standard food.

**EI (early injury) of adult flies using bleomycin or paraquat**

4 d mated female flies (30-50 per vial) were first starved in an empty vial for 2 hours, and then transferred into a vial containing a filter paper

that was soaked with a solution consisting of 25 µg/mL bleomycin or 10 mM paraquat dissolved in 5% sucrose, UT flies were fed with a filter paper that was soaked with 5% sucrose. After 24 h of treatment, flies were transferred to fresh standard food. The midguts of flies were dissected and subjected to immunofluorescent staining at times indicated in the manuscript.

**Fig. 7 | iTRF-induced age mosaic results in alleviated midgut hyperplasia and extended lifespan. a** Schematic of ad libitum (Ad) and iTRF treatment. **b** Time courses of statistics of PH3+ cells per midgut in Ad and iTRF flies. The light blue indicates the time window in which the iTRF fasting is in place. n, numbers of midguts. Ad: n = 14 (10 d), 16 (16 d), 16 (20 d), 14 (24 d), 12 (34 d), 12 (47 d), 14 (57 d). iTRF: 12 (10 d), 20 (16 d), 16 (20 d), 12 (24 d), 14 (34 d), 14 (47 d), 14 (57 d). **c** Upper, design of the puparium H2B-RFP EC labeling and tracing in Ad and iTRF flies. Lower, representative images of the RFP tracing pattern in 18 d, 26 d Ad and iTRF midguts. **d** Statistics of the ratio of RFP+ ECs to total ECs in selected areas of Ad and iTRF posterior midgut during aging. n, numbers of regions. Ad: n = 25 (14 d), n = 22 (18 d), n = 21 (22 d), n = 22 (26 d); iTRF: n = 17 (14 d), n = 20 (18 d), n = 17 (22 d), n = 20 (26 d). **e** Statistics of PH3+ cells in Ad midgut in which *Atg1* was overexpressed in ECs during the night. *attp2*: n = 19, *Atg1*: n = 20. **f** Schematic of *Atg1* or *Atg8a* knockdown in ECs in 30 °C time windows (the iTRF fasting phase) in Ad or iTRF flies.

**g, h** Statistics of PH3+ cells per midgut in Ad and iTRF flies counted at 24 d (**g**) or 70 d (**h**) when *Atg1/Atg8a* was knocked down in the EC within 30 °C time windows (**f**), n, numbers of midguts. n (24 d, from left to right) =18, 20, 20, 19, 19, 20; n (70 d, from left to right) =18, 19, 20, 17, 19, 19. Representative images (**i**) and statistics (**j**) of Dlg1 staining in Ad and iTRF midgut at 35 d. GFP driven by *esg-Gal4* indicates the ISCs/EBs. n, number of ECs measured. 16 (Ad) and 17 (iTRF) midguts are examined. **k** Upper: schematic of the prevention of age mosaic formation by blocking ISC divisions (*esg^ts > Cdk2 RNAi*) in Ad and iTRF flies. Lower: statistics of the PH3+ cells per midgut in control (*attp2*) and age mosaic prevention (*Cdk2 RNAi*) flies on Ad and iTRF treatment at 55 d. n, numbers of midguts. n = 13, 16, 15, 15 (from left to right). **l** Survival analysis of control (*attp2*) and age mosaic prevention (*Cdk2 RNAi*) flies on Ad and iTRF treatment. Data are mean ± SEM. Significance was determined using two-tailed unpaired *t* test. Scale bars, 10 μm (**l**) or 20 μm (**c**). Source data are provided as a Source Data file.

## Induction of EI by early transient EC ablation or early transient ISC proliferation

We adapted the EC ablation or ISC proliferation to induce EI assay to the method described[32]. *UAS-Reaper* was crossed to the temperature sensitive EC-Gal4 line: *Myo1A^ts*. *UAS-Hep^act* and *UAS-Hop^tuml* were crossed to the temperature sensitive ISC/EB-Gal4 line: *esg^ts*. Crosses were reared at 18 °C. Newly eclosed flies were collected to standard food and followed to mate for 48 h. 4 d old mated female flies were transferred to 30 °C for 12 h to activate the Gal4 function to induce the EI, and then returned to 18 °C. The midguts of flies were dissected and subjected to immunofluorescent staining at times indicated in the manuscript.

## Smurf assay

We adapted the Smurf assay to the method described[18]. Standard food with blue dye (Brilliant blue FCF, MedChemExpress, HY-D0915) added at a concentration of 2.5% (wt/vol). Flies were fed this diet for 48 h at 18 °C and for 24 h at 25 °C. A fly was counted as a smurf if dye coloration was observed outside the digestive tract.

## Frequency measurement of Prospero+ tumor

*Notch^55e11/+* flies were cultured at 25 °C. 4 d flies were treated with UT (5% sucrose) or EI (bleomycin-fed) for 24 h, then flies were transferred to fresh standard food. Midguts were dissected at 35 d and subjected to immunofluorescent staining. Prospero+ tumor events were scored as clusters of at least 20 Prospero+ diploid cells.

## Quantification of midgut diameters and midgut area

The flies were cultured at 25 °C. 4 d flies were treated with 5% sucrose (UT) or bleomycin (EI) for 24 h, then flies were transferred to fresh standard cornmeal food. Midguts were dissected in 1x PBS on ice. Images of midguts were taken with a stereoscopic microscope (AXIO Zoom.V16) using the same settings. Average midgut diameters were obtained from the average of 5 measurements in R4C-R5A using ZEN 2.1 (Zeiss stereoscopic microscope software). The area of the midgut was measured using the Curve Contour Area function in ZEN 2.1 confocal software.

## Induction of SI (second injury) in flies that have experienced an EI

For flies that have experienced an EI by bleomycin or paraquat: flies were cultured at 25 °C for 15 days after EI, then flies were treated with *Ecc15* oral infection for 24 h to induce the SI, then the midguts of flies were dissected and subjected to immunofluorescent staining.

For flies that have experienced an EI by transient EC ablation or transient ISC proliferation: flies were reared at 18 °C for 70 days after the EI, then the flies were transferred to 30 °C for 12 h to secondarily activate Gal4 function, then flies were returned to 18 °C. The midguts of flies were dissected and subjected to immunofluorescent staining at times indicated in the manuscript.

## Statistics of the ISC number in midgut that has experienced EI

To calculate the ISC number in *NRE-lacZ; esg-Gal4 > GFP* midgut that has experienced EI, 5 regions of interest in each posterior midgut were imaged for 10 midguts. *NRE-LaZ esg > GFP*+ cells were recognized as ISCs[61] and were calculated in every picture.

## Senescence-associated β-galactosidase assay

For this experiment, we have referred to the method described[62]. The senescence-associated β-galactosidase assay was performed using the Senescence Cells Histochemical Staining Kit (CS0030; Sigma-Aldrich). Briefly, midguts were dissected in 1×PBS on ice, then these midguts were washed twice with 1×PBS (cold) and fixed in 1× fixation buffer for 7 minutes at room temperature. The midguts were then washed three times with 1×PBS and incubated with 1×staining mixture (mixed according to the kit instructions) at 37 °C for 30 minutes. After staining, midguts were washed in 1×PBS and mounted on slides in 1×PBS. Midguts were immediately imaged on a Carl Zeiss LSM 800 confocal microscope using ZEN 2.1 imaging software.

## Gut microbiota 16S rRNA sequencing

Guts commensal bacteria sequencing was performed according to previous reports[21,22]. Untreated (UT) and early injury (EI) midgut bacteria were sequenced in 3-4 independent samples. Each sample contained 20 mated female midguts. Midguts were dissected in 1x sterile PBS solution and DNA was extracted using TIANamp Genomic DNA Kit (TIANGEN Biotech, Cat# DP304-02). The 16 s rRNA sequencing library construction and sequencing process were performed by Majorbio Bio-Pharm Technology Co. Ltd (Shanghai, China). The sequencing results were clustered into operational taxonomic units (OTUs) using UPARSE (version 7.0.1090), and OTUs were clustered for non-repetitive sequences (excluding single sequences) at 97% similarity, with chimeras removed during the clustering process to obtain representative sequences for OTUs. The OTUs were taxonomically annotated using the RDP classifier against the Silva 16S rRNA gene database.

## Colonization of commensal microbes in axenically reared adult flies

We adapted the generation of adult flies carrying a single commensal bacterium (mono-associated flies) assay to the method described[127,128]. Commensal bacterial cultures (~ 10^8 CFUs) were directly added to axenic standard food vials containing germ-free embryos. These vials were then maintained in a sterile cell culture hood before eclosion. After eclosion adult flies were transferred into new, sterile vials every 2 days. The commensal bacteria used in this experiment were: *Acetobacter pomorum* (*Ab*) and *Enterobacter* (*Eb*). To determine the success of colonization, the microbiota status of the adult flies was measured at day 4 post-eclosion: five flies were homogenized in 200 μl deionized 1×PBS, serially diluted to 1 in 10,000 and plated onto the selective plates, the plates were incubated at 30 °C for 2 d.

## Commensal quantification and selective plates

We adapted the commensal quantification assay according to the method described[21,22]. The dissected guts were homogenized in 200 μl sterile PBS and plated onto selective plates, which were incubated at 30 °C for 48-72 h, then quantifies the number of colony-forming units (CFU).

Selective plates were generated according to the following recipes: Acetobacteriaceae: 25 g/L D-mannitol, 5 g/L yeast extract, 3 g/L peptone, and 15 g/L agar. Enterobacteriaceae: 10 g/L Tryptone, 1.5 g/L yeast extract, 10 g/L glucose, 5 g/L sodium chloride, 12 g/L agar.

Nutrient Rich Broth: 23 g/L BD Difco Nutrient agar.

All media were autoclaved at 121 °C for 20 min.

## Axenic fly culture

Sterile food in vials was prepared by autoclaving at 121 °C for 30 min and cooled in the sterile cell culture hood. Flies were sterilized and reared under sterile conditions according to previous reports[21,128]. Briefly, the embryos collected from the agar plate were washed in Walch's solution (1:30 dilution) for 3 times, then the eggs were bleached in 2.7% sodium hypochlorite for once and 70% ethanol for twice, and then washed twice with sterile ddH2O. These embryos were then transferred to vials containing sterile food. These vials were maintained in a sterile cell culture hood before eclosion. After eclosion, adult flies were transferred to new sterile vials every 2 days in the sterile cell culture hood.

## The daily time-course RNA-sequencing and data analysis

Prior to the experiment, 64 bottles of *esg-Gal4 UAS-GFP* flies were prepared for the same aging fly collection. 3000 newly eclosed female flies were collected in 24 hours. Each 50-60 females with 5-10 males were reared in bottles with fresh cornmeal food changed every 2 days.

Half of the flies were fed bleomycin for 24 h at 4 d (EI) and the other half were fed 5% sucrose (UT). Flies were then transferred to new bottles with fresh cornmeal food. 3 samples of 4 d flies, each sample containing 20 female flies, were dissected as the samples at the start time point. 3 samples of 6 d UT and 3 samples of 6 d EI flies were dissected as the post-EI time point sample. From 12 d to 28 d, 3 samples of UT and 3 samples of EI flies were dissected each day to obtain a daily time course transcriptome of the aging process.

Midguts were dissected in RNA*later* solution (Invitrogen, life), the dissected midguts were then immediately frozen on dry ice and used for total RNA preparation by isothiocyanate-alcohol phenyl-chloroform. Total RNA was extracted from 20 female midguts for RNA-seq. RNA integrity was assessed using the RNA Nano 6000 Assay Kit of the Bioanalyzer 2100 system (Agilent Technologies, CA, USA). Library construction was performed using the NEB Next Ultra II DNA library prep kits (New England Biolabs, cat.no. E7645L) and sequencing were performed by Novogene Technology (China). Briefly, library was performed and sequenced on an Illumina Novaseq platform and 150 bp paired-end reads were generated. Raw data in fastq format were first processed using fastp software. Reads were aligned to the *Drosophila* reference genome (BDGP6) using hisat 2.0.5[129]. Mapped reads from each sample were assembled using StringTie (v1.3.3b) in a reference-based approach. Differential expression gene analysis was performed using DESeq Rpackage (1.20.0)[130]. The differentially expressed genes (DEGs) were identified with edgeR (4.0.16) and the log2 fold change shrinkage method was applied. DEseq2 was employed to identify significantly differentially expressed genes with the following parameters: *p*-value < 0.05 and the absolute value of Fold change>2. Gene ontology (GO) enrichment analysis of DEGs was performed using Metascape.

## Tracing EC turnover using H2B-RFP

We adapted the labelling of ECs using the long-lived H2B-RFP assay according to the method described[30]. To trace the EC turnover in aged midgut that has experienced EI: *Myo1A^{ts}/ Mex^{ts} > H2B-RFP* flies were cultured at 18 °C, 4 d old flies were treated with 5% sucrose (UT) or bleomycin (EI) for 24 h, then flies were transferred into fresh standard food. When the flies were 44 days old, the flies were transferred to 30 °C for 24 h to label ECs by transient expression of H2B-RFP, and then returned to 18 °C.

To label and trace the puparium generated ECs, *Myo1A^{ts}/ Mex^{ts} > H2B-RFP* flies were cultured at 18 °C. When pupa grew to 72 h APF, flies were transferred to 30 °C for 24 h to label ECs by transient expression H2B-RFP, then flies were returned to 18 °C. Newly eclosed flies were collected and mated for 48 h, then 4 d old flies were treated with 5% sucrose (UT) or bleomycin (EI) for 24 h, and then flies returned to 18 °C on fresh standard food.

To label and induce the EC turnover at the same time, *Myo1A^{ts} > UAS-Reaper & UAS-H2B-RFP* flies were cultured at 18 °C, newly eclosed flies were collected and mated for 48 h, then 4 d old flies were transferred to 30 °C for 12 h to activate the Gal4 function to label ECs and ablate part of ECs at the same time, then flies were returned to 18 °C on fresh standard food.

The midguts of flies were dissected and subjected to immuno-fluorescent staining at times indicated in the manuscript. Images were taken in the R4C-R5A regions[131] of the *Drosophila* posterior midgut using a 20× objective.

## *upd3*-LacZ, *vn*-LacZ, HP1, STAT-GFP and *gstd1*-GFP fluorescence quantification

Flies in the control and experimental groups were dissected, fixed and immunostained under the same conditions. Confocal images were captured using the same settings. Cells with large polyploid nuclei were recognized as ECs. Fluorescence intensity of regions of interest (ROI) was measured using the mean gray value function in the confocal software ZEN 2.1, and then subtracting the mean fluorescent background calculated from 3-5 ROI neighboring blank areas.

## Total RNA extraction and quantitative reverse transcription PCR (qRT-PCR)

Total RNA was extracted from the midgut of 20 female flies per sample using the RNAprep Pure Tissue Kit (TIANGEN Biotech, Cat# DP431). The GoScript™ Reverse Transcription Mix, Oligo(dT) Kit (Promega, Cat# A2790) was used to generate cDNA from a total of 500 ng RNA. qRT-PCR was performed in 3 independent biological replicates using GoTaq® qPCR System (Promega, Cat# A6001). Expression values were calculated using the ΔΔCt method and relative expression of target genes was normalized to the mean of the reference genes *RpL23*. The expression in 5 d UT sample was further normalized to 1. Results of mRNA expression obtained by qPCR are presented as mean± s.e.m of 3 independent biological samples.

Primer sequences:

*RpL23_F:* GACAACACCGGAGCCAAGAACC
*RpL23_R:* GTTTGCGCTGCCGAATAACCAC
*upd3*_F: GAGCACCAAGACTCTGGACA
*upd3*_R: CCAGTGCAACTTGATGTTGC
*vn*_F: GAACGCAGAGGTCACGAAGA
*vn*_R: GAGCGCACTATTAGCTCGGA
*socs36E*_F: CAGTCAGCAATATGTTGTCG
*socs36E*_R: ACTTGCAGCATCGTCGCTTC

## Quantification of Lamin, LamC, Dlg1, Cora and E-cadherin-GFP immunofluorescent staining

Flies in the control and experimental groups were dissected, fixed and immunostained under the same conditions. Confocal images were captured using the same settings. Fluorescence intensity of regions of interest (ROI) was measured using the mean gray value function in the confocal software ZEN 2.1, and then subtracting the mean fluorescent background calculated from 3-5 ROI neighboring blank areas. The SJ or AJ to cytoplasmic (cyto) fluorescence ratio was obtained by dividing

the average membrane fluorescence intensity by the average cytoplasmic intensity. 3 confocal images were taken from each posterior midgut, and at least 14 midguts per group were analyzed.

### In vivo detection of ROS

In vivo ROS detection was performed in live tissue using either dihydroethidium (DHE, Invitrogen, D23806) or 2′,7′-Dichlorodihydrofluorescein diacetate (H2DCF, MedChemExpress, HY-D0940) as described[23,88,89]. For the DHE staining, flies were dissected in the Schneider's insect medium on ice. Whole tissues were then incubated with 60 μM DHE for 7 minutes on a shaker at the room temperature in the dark, the midguts were then washed three times with Schneider's medium and once in 1×PBS for 5 min on a shaker in the dark. For the H2DCF staining, midguts were dissected in 1×PBS on ice. Midguts were then incubated with 20 μM H2DCF on a shaker at room temperature for 25 min in the dark. Then midguts were washed three times in 1×PBS for 5 min. After staining, midguts were mounted on slides in 1×PBS. Images were captured immediately using confocal microscope. Fluorescence intensity of regions of interest (ROI) was measured using the mean gray value function in the confocal software ZEN 2.1, and then subtracting the mean fluorescent background calculated from 3-5 ROI neighboring blank areas.

### Antioxidant feeding

N-Acetyl Cysteine (NAC) (MedChemExpress, HY-B0215) was diluted in H2O and added to the cornmeal-agar food at a final concentration of 20 mM. 4 d old mated female flies were reared on this food, and transferred to fresh NAC food every 2 days. Midguts were dissected and subjected to immunofluorescent staining at times indicated in the manuscript.

### Knockdown of CncC, Lamin and LamC in ECs

Control (UAS-attp2) and UAS-Lamin RNAi, UAS-LamC RNAi and UAS-CncC RNAi were crossed to the temperature sensitive EC-Gal4 lines. Crosses were reared at 18 °C. Newly eclosed F1 flies were collected and transferred to 30 °C at 4 d. After 7 days at 30 °C, Lamin and LamC knockdown midguts were dissected and subjected to immunofluorescent staining, and after 12 days at 30 °C, CncC knockdown midguts were dissected and subjected to immunofluorescent staining.

### Knockdown of Dlg1, Mesh, Cora and Tsp2A in ECs that have experienced an EI

Control (attp2) and UAS-Dlg1 RNAi, UAS-Mesh RNAi, UAS-Cora RNAi and UAS-Tsp2A RNAi were crossed to Myo1A[ts]. Crosses were reared at 18 °C. 4 d adult female flies were treated with bleomycin for 24 h to induce an EI, then the flies were transferred to fresh standard food to recover for 6 d. Flies were then transferred to 30 °C for 6 d to activate the Gal4 function, midguts were dissected and subjected to immunofluorescent staining.

### Temporarily halting (TH) the ISC division in hyperplastic midgut

Control (attp2) and esg[ts] > UAS-Cdk2 RNAi flies were reared at 18 °C. 65 d old flies with midgut hyperplasia were transferred to 30 °C for 7 days, which temporarily halted ISC divisions. Flies were then returned to 18 °C with fresh standard food. Midguts were dissected and subjected to immunofluorescent staining at times indicated in the manuscript.

### iTRF assay

We adapted the iTRF assay to the method described[101]. Newly enclosed adult female flies were collected within 24 hours. 10 d old flies were fasted for 20 h every other day, starting at mid-morning [6 h after lights on or Zeitgeber time 6 (ZT6)], with 28 h recovery of ad libitum (Ad) feeding between fasting days. The iTRF treatment was carried out for a 20-day window from 10 d to 30 d. Then files were reared on

standard food for Ad feeding. The fasting media consisted of 1% agar in ddH2O, which was freshly prepared daily.

### Preventing the formation of age mosaic during iTRF treatment

Control (attp2) and esg[ts] > UAS-Cdk2 RNAi flies were reared at 18 °C. When 10 d flies began the iTRF fasting cycles, flies were transferred to 30 °C to block the ISC division preventing the formation of EC age mosaic. After the iTRF treatment, flies were reared at 18 °C on Ad diet. Midguts were dissected and subjected to immunofluorescent staining at times indicated in the manuscript.

### Night-biased ectopic expression of Atg1 in ECs

Control (attp2) and UAS-Atg1 were ectopically expressed using the flowing temperature sensitive EC-Gal4 line: Myo1A[ts]. Crosses were reared at 18 °C. Flies were reared on fresh standard food with a 12-h light-dark cycle. 10 d flies were transferred to 30 °C from ZT12 to ZT24 to activate the Gal4 function, then returned to 18 °C from ZT24 to ZT0. After repeating these treatments for 6 days, midguts were dissected and subjected to immunofluorescent staining.

### Knockdown components of autophagy in ECs during the iTRF fasting phase

Control (attp2) and Myo1A[ts] > UAS-Atg1 RNAi or UAS-Atg8a RNAi flies were reared at 18 °C. When 10 d flies began the iTRF fasting cycles, flies were transferred to 30 °C during the night fasting period to block the autophagy. Except for the fasting period, flies were reared at 18 °C on Ad diet. Midguts were dissected and subjected to immunofluorescent staining at times indicated in the manuscript.

### Chloroquine (CQ) feeding during the iTRF fasting phase

Chloroquine (CQ) was added to 1% agar fasting media at a final concentration of 2.5 mg/mL[132]. To block the autophagy during the iTRF fasting phase, flies were fed on fasting media with CQ. Except for the fasting period, flies were transferred to standard food for Ad feeding.

### Lifespan analysis

To measure the lifespan of adult flies, female flies were collected within 24 hours of eclosion. Flies were transferred to plastic cages (175 mL volume, 5 cm diameter, approximately 60 females per cage with several males) with standard cornmeal food and reared at the temperature indicated in the manuscript. Food was changed every 2 days, and the number of dead females was visually identified and recorded each day until all females died. Lifespan data were analyzed using Prism statistical software.

### Immunostaining and fluorescent microscopy

The midguts of flies were dissected in 1×PBS (Solarbio, Cat# P1010), and fixed in 4% formaldehyde (Sigma, Cat# F8775) for 3 h, then washed and permeabilized three times for 20 min with 0.3% PBT (1×PBS solution containing 0.3% Triton X-100 (Sangon Biotech, Cat# A110694-0500)). The samples were then incubated in primary antibody for 3 hours at room temperature with shaking and washed three times for 20 minutes with 0.3% PBT. Lastly, samples were incubated with 100 μL DAPI (1 μg/mL, Sigma, Cat# D9542) for 10 min, washed three times for 30 min with 0.3% PBT and mounted in 70% glycerol (Sinopharm Chemical Reagent, Cat# 10010618). Primary antibodies were the following and used at the following dilutions: chicken anti-GFP at 1:10,000 (Abcam, Ab13970); rabbit anti-GFP at 1:10,000 (Abcam, Ab6556); rabbit anti-GFP (1:10,000, Abcam, Cat# AB290), mouse anti-Pros (1:200, Developmental Studies Hybridoma Bank, Cat# 528440), chicken anti-β-galactosidase at 1:10,000 (Abcam, Ab9361), rabbit anti-PH3 (1:10000, Millipore, Cat# MMI-06-507), mouse anti-Discs large 1 (1:100, DSHB, Cat# 4F3), mouse anti-Armadillo (1:100, DSHB, Cat# N2 7A1), mouse anti-HP1(1:100, DSHB, Cat# C1A9) mouse anti-Lamin (1:100, DSHB, Cat# ADL84.12), mouse anti-LamC (a kind gift from

Haiyang Chen, Sichuan Universtiy,1:100, DSHB, Cat# LC28.26)[94], mouse anti-Cora (1:100, DSHB, Cat# C615.16). The following secondary antibodies were used: Alexa Flour goat anti-chicken 488 (1:4000, Invitrogen, Cat# A11039), Alexa Flour goat anti-rabbit 488 (1:4000, Invitrogen, Cat# A11008), Alexa Flour goat anti-rabbit 555 (1:4000, Invitrogen, Cat# A21428), Alexa Flour goat anti-mouse 555 (1:4000, Invitrogen, Cat# A21422), Alexa Flour goat anti-mouse 647 (1:4000, Invitrogen, Cat# A21235). Midguts were imaged with a confocal microscope (Zeiss LSM800) using Plan-Apochromat 20x/0.8 M27, 63x/1.40 Oil DIC M27 objectives (imaging medium: immersol™ 518 F obtained from Zeiss). The acquisition and processing software used was ZEN 2.1. Images were processed in Photoshop CC (Adode) and Illustrator CS6 (Adobe) for image merging and resizing.

## Statistical analysis
Statistical analysis was performed using Prism (GraphPad Software). Unpaired student t-test was used to compare two groups. Results were expressed as mean ± s.e.m. Significance was accepted at *, $p < 0.05$; **, $p < 0.01$; ***, $p < 0.001$ and ****, $p < 0.0001$. All the statistical details of experiments can be found in the figures and figure legends.

## Reporting summary
Further information on research design is available in the Nature Portfolio Reporting Summary linked to this article.

## Data availability
All data generated or analyzed during this study are available as a Source data flies. The raw daily RNA-seq sequence data and gut microbiota sequencing data reported in this paper have been deposited in the Genome Sequence Archive[133] in National Genomics Data Center[134], China National Center for Bioinformation/Beijing Institute of Genomics, Chinese Academy of Sciences (GSA: CRA016935 and GSA: CRA027274, CRA027275) that are publicly accessible at: https://ngdc.cncb.ac.cn/gsa. Drosophila stocks and reagents generated in this study are available upon request to Z.G. (guozheng@hust.edu.cn). Source data are provided with this paper.

## Code availability
The code for analyzing the daily RNA-seq has been deposited in the Figshare database without accession code that is publicly accessible at: https://figshare.com/account/home#/projects/220879.

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

## Acknowledgements

We thank HY. Chen, ZZ. Zhai, WZ. Ge, HP. Yu, BDSC, VDRC, DGRC, NIG-Fly and Tsinghua Fly Center for fly strains; HY. Chen and DSHB for antibodies. QP. Wang for bacteria strains. LY. Zhang and YY. Han for critical comments and insightful suggestions. This work was supported by grants from the National Natural Science Foundation of China to Z.G. (31771625, 31970817, 32271074, 32470885).

## Author contributions

P.Q. and Z.G. conceptualized and designed experiments. P.Q. and Y.W. performed the bulk RNA-Seq. Y.W. performed bioinformatics analysis of the RNA-seq data. P.Q., Q.W., Q.Y., M.L., and Z.G. performed experiments and interpreted data; P.Q. and Z.G. wrote the manuscript. Z.G. supplied resource and funding.

## Competing interests

The authors declare no competing interests.
