## [Transparent Peer Review file · Nature Communications]

Age mosaic of gut epithelial cells prevents aging

Corresponding Author: Professor Zheng Guo

Version 0:

Reviewer comments:

Reviewer #1

(Remarks to the Author)

In this manuscript, Qin et al., describe ageing that early injury introduces new EC into the midgut epithelium for the benefit of preventing aging hyperplasia by creating a genetic mosaic of EC. Interestingly, intermittent fasting recreated these effects and might serve as a therapeutic approach. Especially the finding of massive EC death suggesting a fixed lifespan for EC is intriguing. The authors perform a series of genetic and pharmacological experiments to support their claims. Their findings are novel and represent an important leap towards better understanding of intestinal hyperplasia connected with ageing, which justify publication in Nature Communications.

Although the authors provide phenotypical and statistical evidence for their conclusions, I have major conceptual and experimental considerations and minor remarks that have to be addressed during a revision.

Major

1. In Fig.1 the authors show that EI markedly improves fly survival, which is definitely counterintuitive at first sight. To my understanding, Myo driven reaper should basically kill all EC in 12-24h in their experiments. Why should these EC not be synchronized then in a way that the hyperplasia is not just postponed?
2. In the same line asking about the extend of Myo>reaper and EI: is there really a mosaic formed in many flies upon EI? I think this hypothesis should be underlined with data from clonal (O'Brien lab, 2017) and/or dual expression ReDDM/Rapport (Reiff lab, 2024 research square preprint) tracing analysis. Such experiments will allow direct assessment and visualization of mosaic patches or full midgut renewal. pH3 and esg counts are a good hint, but not necessarily connected to renewal of patches or full population of EC as they remain untraced.
3. The authors did very thorough and convincing work on whether the microbiota contribute to EI. At least equally important are data points whether after EI, full gut size/cell number is restored to previous size and to include the extend of EI reduction (point#2):
 - a. What happens to gut diameter and length upon and after EI?
 - b. If reduced, are fertility and fecundity as general measure for intact physiology affected?The authors claim their findings to be of therapeutical value, thus it is vital to sustain this statement to control for these phenomena shown by the Dominguez, Edgar, Miguel-Aliaga and Reiff labs.
4. In the same line, in their material and methods the authors also state that only mated female flies were used in the experiments. What does this exactly mean: experiments have been done in mated females with or without males in the vial during the experiment? This is a crucial point and in case it was unclear, key experiments such as Figure 1 need to be done comparing, once mated and constantly mated females. In first case, it also needs to be clear whether females were 'mated' long enough to exclude having remained virgins throughout the experiment as this strongly affects survival.
5. A sub-concern of this: day 20 might mark the day, when isolated UT and EI female flies either run out of stored sperms or enter menopause-like states. What happens to egg laying after EI and day 20?
6. In the same line as points 3 + 4: it is only two representative images, but gut diameters of EI flies in Fig3a and 3i seem reduced compared to UT flies. Diameter measurements would be helpful for interpretation here.
7. Why EI spans 12h in some experiments and 24h in others? That should impact EC numbers by a factor of 2.
8. The authors need to provide data on Myo> expression in other tissues to exclude gross physiological effects.

9. At least some key experiments (Fig1) should be confirmed using Mex-Gal4 which shows more uniform activation patterns compared to Myo-Gal4.

10. Fig.6g states that new EC improve the SJ Dlg1 staining with old EC. This phrase ('indicating that the new ECs improved the SJs of the old ECs surrounding them.') is very speculative as no expansion or high-resolution microscopy was performed that would show from which of the two cell membranes the stronger signal originates. It could very well be that all additional signal intensity derives from the new EC. For this, in the same samples, SJ of old EC contacting each other could be measured vs mixed vs new-new EC.

11. Functionality of RNAi stocks must be validated. Either by citing papers that validated knockdown efficiency of particular stocks, qPCR or at least reproduction with a second stock targeting the same mRNA.

The idea of reverse 'ReDDM' labelling the EC population is really useful (and cool). It also shows that the authors use high quality fly food stimulating low turnover in combination with flipping every 48h (M&M), which allows observation in homeostasis. The authors should also consider discussing that hyperplastic cells remain as their homeostatic midgut turnover is really low.

Regarding this method, the cartoons in Fig.3+4 sum up data that is not shown in the confocal images and thus are an assumption. *esg+* are not addressed so it is not formally proven that these guts have hyperplasia. This needs to be shown directly (*esg*-GFP from Leanne Jones lab or *armadillo/pros* stainings) or removed.

Minor

12. Regarding the data of Fig.7: The authors may consider to reveal the full extent of the EI/iTRF effect with survival curves with ITRF intervention at 60d to reveal whether iTRF is able to extend life span when hyperplasia already happened and 'help' old individuals.

13. In the same line: Conclusions could be strengthened if *Myo*> forcing *Atg1/8a* expression in a *la* EI would extend lifespan.

14. For gene description/protein labelling please refer to Flybase nomenclature (e.g. Dlg1 instead of Dlg) throughout the manuscript.

15. It should also be stated clearly where in the midgut these measurements were performed. Regions differ vastly and investigations should only focus on one region such as R5.

16. The role of Lamins in the *Drosophila* midgut was investigated by Joerg Großhans lab before and should be cited accordingly.

Reviewer #2

(Remarks to the Author)

In this article titled "Age mosaic of gut epithelial cells prevents aging", the authors have explored the idea of healthy aging by inducing early gut epithelial renewal in young organisms. They have performed extensive transcriptomic analyses of aging *Drosophila* midguts. This data set supports the idea that early injuries to epithelial lining is beneficiary to rejuvenate the gut, thereby increasing the lifespan. Even though the idea of early injury seems interesting, this reviewer is not completely sure how relevant it is in a model like mammalian gut which undergoes a turnover every 3-4 days for several years whereas *Drosophila* has a turnover of 2-3 weeks for few months only. The generation of new ECs by TA cells surrounding the villi is by itself providing this event. This reviewer thinks that their idea of pushing this approach towards therapeutic purpose could be a bit of an overstatement.

Major Comments:

1. The main idea that authors propose is that in aged guts, old enterocytes (ECs) undergo collective turnover, leading to hyperplasia. When they say that ECs undergo collective turnover, do they mean that ECs are lost all at once or still lost in a controlled fashion? Can the authors look for markers for cell death in ECs of aged midgut to get an idea of the dynamics of epithelial removal? Could the authors show some of the pictures including the green cells (ISC-EB) in hyperplasia before to only use cartoons.

The strategy they use for inducing early injury (EI) is by either genetic means by expressing Reaper in ECs or using pharmacological stressors. These treatments can also induce an uncontrolled collective turnover of ECs but seem not to induce hyperplasia when the guts are left to recover. Does this mean that the way a midgut recovers depends also on the age of the organism? What is the extent of damage to ECs by transient expression of Reaper, bleomycin, and paraquat? The way pharmacological treatments (bleomycin or paraquat) are used in this study on their own has a toxic effect. Could the authors optimize the exposure time and dosage of these treatments so that it is enough to generate a turnover of epithelial cells but won't have an overall toxic effect?

2. This reviewer finds it difficult to understand why flies that are of different ages are classified as old. For example, in Figure 1B, there is data from flies that are 100 and 150 days old whereas in Figure 1C or 1I, there is data from only till 40 or 21 days. Are these flies old or are they stressed? Have the authors looked for some markers of aging to be sure that these are

old flies? Is this difference due to the treatment itself or genetic background? A survivorship curve for these conditions like in Figure 1A would help the reader understand the choice of days of age for classifying as old flies.

3. In Figure 3, the authors conclude that EI leads to a slowdown of EC turnover in the aged midgut. It is known that ISC hyperproliferation can accelerate EC turnover and aged UT midguts have hyperplasia. In contrast, EI midguts do not exhibit hyperplasia. Could the accelerated EC turnover in aged UT midguts be an outcome of hyperplasia? Can the authors block ISC proliferation and follow EC turnover in the aged UT midguts?

4. Authors propose the idea of accumulation of ROS in old ECs as an upstream signal for collective EC turnover. Can the authors test this directly by reducing the intracellular ROS maybe by feeding the flies with ROS scavenger like N-Acetyl Cysteine and then check EC turnover in UT flies upon aging? Can the authors also check the EC turnover using their technique upon knocking down CnC to increase cellular ROS?

Interestingly, from Figure 5d, it is not clear if all old ECs in 30D Ut midguts are in high ROS state. Can the authors plot the mean intensity of GFP in RFP+ ECs between 15D UT and 30D UT conditions?

5. Is inhibition of collective EC turnover sufficient to inhibit hyperplasia? Can the authors express a blocker of cell death like p35 in ECs to see if this is sufficient to block hyperplasia?

6. The authors suggest that the septate junctions (SJs) in old ECs are rescued when they are neighboring a new EC (see model 6g). But in Figure 6b or e, SJs are well restored in old ECs of EI midguts irrespective of whether they neighbor a new EC or old EC. If disrupted SJs are the driving force for collective EC turnover, I am puzzled by finding large clones of old ECs with disrupted SJs in 15D EI midguts. Can the authors test if restoring the expression of SJ components in old ECs is sufficient to block collective EC turnover and hyperplasia?

Minor Comments:

1. How does the Lamin distribution in the newly produced ECs in 50D UT midgut look like? Is restoring Lamin sufficient in old ECs to prevent their collective turnover?

2. Similarly, in old UT midguts (30D for example), there is still a bit of EC turnover. When these new ECs form in between old ECs, does this rescue the septate junctions (SJs) in old ECs?

3. In general, it would be great if the confocal micrographs are not too cropped for example, like in Figure 5d, g, 6b etc. I prefer to see a bigger overview of the midgut region so that the mosaicity of old and new ECs is clearer.

Reviewer #3

(Remarks to the Author)

Version 1:

Reviewer comments:

Reviewer #1

(Remarks to the Author)

Dear authors,
dear editor,

during the revision process, the authors addressed all my (and the other reviewers) concerns extensively and significantly improved the manuscript, which I hope to read in NComms soon.

Finally, I want to congratulate the authors for their findings and will be happy to review more of their work in the future.

Reviewer #2

(Remarks to the Author)

I think that the authors have adequately addressed my comments in the revised version of the manuscript. Therefore, I have no further comments.

Reviewer #3

(Remarks to the Author)

We would like to thank all the Reviewers for their kind words, complimenting the significance and technical quality of our work, and for their constructive suggestions for improvement. As a result of addressing these comments, we believe that our manuscript has improved significantly. Below we provide a point-by-point response (in blue) to the Reviewers' feedback.

Reviewer #1 (Remarks to the Author):

In this manuscript, Qin et al., describe ageing that early injury introduces new EC into the midgut epithelium for the benefit of preventing aging hyperplasia by creating a genetic mosaic of EC. Interestingly, intermittent fasting recreated these effects and might serve as a therapeutic approach. Especially the finding of massive EC death suggesting a fixed lifespan for EC is intriguing. The authors perform a series of genetic and pharmacological experiments to support their claims. Their findings are novel and represent an important leap towards better understanding of intestinal hyperplasia connected with ageing, which justify publication in Nature Communications.

Although the authors provide phenotypical and statistical evidence for their conclusions, I have major conceptual and experimental considerations and minor remarks that have to be addressed during a revision.

We thank the reviewer for the positive evaluation of our work and appreciate the constructive comments and suggestions that helped improving our manuscript.

Major

1. In Fig.1 the authors show that EI markedly improves fly survival, which is definitely counterintuitive at first sight. To my understanding, Myo driven reaper should basically kill all EC in 12-24h in their experiments. Why should these EC not be synchronized then in a way that the hyperplasia is not just postponed?

We thank the Reviewer for highlighting this important point. Indeed, we have tested different durations using *Myo1A^{ts}* (*Myo1A-Gal4 tub-Gal80^{ts}*) to express Reaper, and 12 hours is an optimal injury condition to prevent aging hyperplasia. By transferring *Myo1A^{ts}>UAS-Reaper* from 18°C to 30°C for 12 hours, this transient Reaper expression cannot induce a synchronized EC cell death, which could be directly observed from our Fig. 4h. When we co-expressed H2B-RFP and Reaper by *Myo1A^{ts}* for 12 hours to label ECs and induce EC apoptosis, progressive mosaic cell death could be observed on an 8-day time scale. We speculated that transferring *Myo1A^{ts}>UAS-Reaper* flies from 18°C to 30°C for 12 h induced uneven, low levels of Reaper expression in ECs, resulting in heterogeneous apoptosis.

Fig. 4h. The tracing pattern of H2B-RFP⁺ ECs in UT and *Myo1A^{ts}>Reaper* midguts. *Myo1A^{ts}>Reaper* flies were reared at 18°C and were transiently transferred to 30°C for 12h at 4d to induce early injury (EI).

2. In the same line asking about the extend of Myo>reaper and EI: is there really a mosaic formed in many flies upon EI? I think this hypothesis should be underlined with data from clonal (O'Brien lab, 2017) and/or dual expression ReDDM/Rapport (Reiff lab, 2024 research square preprint) tracing analysis. Such experiments will allow direct assessment and visualization of mosaic patches or full midgut renewal. pH3 and esg counts are a good hint, but not necessarily connected to renewal of patches or full population of EC as they remain untraced.

Thanks to the Reviewer for these valuable suggestions. As shown in Fig. 4h, inspired by ReDDM (we have cited the original ReDDM study¹ and the latest ReDDM/Rapport study published by Reiff lab in Nature Communications²), we used the *Myo1A^{ts}* to transiently express H2B-RFP by transferring flies to 30°C for 12 hours, labelling all ECs, and then rearing flies at 18°C, so that when EI leads to EC turnover, the newly produced ECs were not labelled with H2B-RFP. This method allows us to directly visualize the mosaic patches formed by the old and new ECs.

3. The authors did very thorough and convincing work on whether the microbiota contribute to EI. At least equally important are data points whether after EI, full gut size/cell number is restored to previous size and to include the extend of EI reduction (point#2):

a. What happens to gut diameter and length upon and after EI?

Thanks to the Reviewer for this important question. We measured the average diameter of the R4C-R5A region and the whole gut area of UT and EI flies at 6d and 12d. Our bleomycin feeding injury was performed at 4d for 24 hours, so 6d is the upon EI and 12d is the after EI. Our data shows that upon EI, there was a slight but significant increase in gut diameter and area (Extended Data Fig. 1g, h). However, after recovery from EI, both diameters and gut areas were restored to the UT group (Extended Data Fig. 1g, i). We have added these information in the main text and in the Extended data.

Extended data Fig. 1g-i

g, Representative image of the midgut of UT and EI flies at 6d and 12d. **h, i**, Quantification of the average diameter of the R4C-R5A region of the midgut and the midgut area in UT and EI flies. Data are mean \pm SEM. Significance was determined using two-tailed unpaired t test. Scale bars, 200 μ m.

b. If reduced, are fertility and fecundity as general measure for intact physiology affected?
 The authors claim their findings to be of therapeutical value, thus it is vital to sustain this statement to control for these phenomena shown by the Dominguez, Edgar, Miguel-Aliaga and Reiff labs.

Thanks to the Reviewer for this important question and valuable suggestions. As shown in the Extended data Fig. 1g-i, EI did not reduce the diameter and area of the midgut. Nevertheless, to measure the fertility and fecundity that EI might affect, we counted the egg laying within an aging cohort. Newly eclosed females were collected and raised in cornmeal vials. We crossed 3 2–3 day old females with 3 young males in a new standard cornmeal vial or in a new cornmeal vial with yeast paste (yeast food). Laid eggs from 16 UT vials and 19 EI vials on normal food, and 19 UT vials and 17 EI vials on yeast food were collected and counted each day. We found that after bleomycin feeding, EI reduced egg laying over the next few days compared to the UT group, and as the injury recovered, egg laying in the EI group was no longer significantly different from that of the UT group, but in aged flies, egg laying in the EI group was again significantly less than that of the UT group on both cornmeal and yeast diets (Response Fig. 1), suggesting that bleomycin-fed

EI caused damage to the reproductive system of female flies.

Although we believe that age mosaic of the intestinal epithelium improves the intestinal barrier, avoids hyperplasia and extends the lifespan of *Drosophila*, our data suggest that chemical damage such as bleomycin has deleterious effects on the reproductive system. That's why we started looking for a way to physiologically generate EC age mosaic from 2021 onwards, and finally found that iTRF diet could lead to age mosaic. Due to the time constraints of the revision, we could not further verify whether iTRF is beneficial for fertility in the aged female flies, this could be an interesting topic and we will investigate this in the future studies.

Response Fig. 1. Egg laying numbers by 3 females per day along with age on standard cornmeal food or yeast food. Flies were fed bleomycin for 24h at 4d to induce EI. Data are mean ± SEM. Significance was determined using two-tailed unpaired t test.

4. In the same line, in their material and methods the authors also state that only mated female flies were used in the experiments. What does this exactly mean: experiments have been done in mated females with or without males in the vial during the experiment? This is a crucial point and in case it was unclear, key experiments such as Figure 1 need to be done comparing, once mated and constantly mated females. In first case, it also needs to be clear whether females were 'mated' long enough to exclude having remained virgins throughout the experiment as this strongly affects survival.

We thank the Reviewer for pointing this important issue out. We understand the importance of mating status on female fly physiology and midgut homeostasis³. Therefore, "only mated female flies were used" doesn't mean that we cultured flies with only "mated females" in vials or bottles. For vial culture, typically 15-20 female flies were co-cultured with 5-8 young males, and for bottle culture, 50-60 female flies were co-cultured with 15-20 young males. We have included this information in the Methods section as follows (cited literatures from Bruce Edgar lab and Tobias Reiff lab)^{3,4}: *15-20 female flies were co-cultured with 5-8 young males in vials and 50-60 female flies were co-cultured with 15-20 young males in bottles.*

5. A sub-concern of this: day 20 might mark the day, when isolated UT and EI female flies either run out of stored sperms or enter menopause-like states. What happens to egg laying after EI and day 20?

We thank the Reviewer for this interesting question. We co-cultured female flies with males,

so we supposed that the sperm is not a limiting factor for egg laying. The egg laying after EI and day 20 is shown in Response Fig.1.

6. In the same line as points 3 + 4: it is only two representative images, but gut diameters of EI flies in Fig3a and 3i seem reduced compared to UT flies. Diameter measurements would be helpful for interpretation here.

We thank the Reviewer for their insightful suggestions. Following suggestions, we compared the diameter of the R4C-R5A region and the area of the whole midgut between UT and EI at 25°C at 22 days (aged). Indeed, we found that the diameters and areas of the EI midgut were reduced compared to the UT midgut (Response Fig. 2a-c). We speculate that hyperplasia caused by ISC hyperproliferation produces more epithelial cells, leading to an expansion of the intestinal area, and that EI midgut avoids hyperplasia, leading to a reduction in epithelial area in aged flies.

Response Fig. 2a-c

a, Representative image of the midgut of UT and EI flies at 22d. b, c, Quantification of the average diameter of the R4C-R5A region of the midgut and the midgut area in UT and EI flies. Data are mean \pm SEM. Significance was determined using two-tailed unpaired t test. Scale bars, 200 μm .

7. Why EI spans 12h in some experiments and 24h in others? That should impact EC numbers by a factor of 2.

We thank the Reviewer for pointing this out. The choice of 12 or 24 h EI treatment was based on whether it was a genetic manipulation (12 h) or a drug treatment (24 h). For the use of the temperature-sensitive *tub-Gal80^{ts}*, *Gal4/UAS* expression system, we found that transferring 18°C-fed *Drosophila* to 30°C for 12 h was optimal to activate the expression of downstream genes of the UAS, which could stably generate an EC age mosaic. Regarding the use of drug feeding (bleomycin or paraquat) to generate EI, we tried different feeding durations and found that 24 h feeding could stably generate an EC age mosaic.

8. The authors need to provide data on Myo> expression in other tissues to exclude gross physiological effects.

We thank the Reviewer for this suggestion. We used *Myo1A-Gal4^{ts}* to drive the *UAS-GFP* for 12 h to visualize the GFP expression pattern in the midgut/brain/ovary. We found that in addition to EC expression in the midgut, a few GFP⁺ cells were observed in the brain (Response Fig. 3). To rule out physiological effects of *Myo1A-Gal4* in these brain cells, we repeated some key experiments using the *Mex-Gal4 tub-Gal80^{ts}* system (*Mex^{ts}*) as suggested by the reviewer below.

Response Fig. 3. Expression of *Myo1A-Gal4>GFP* in the midgut, the brain and the ovary.

9. At least some key experiments (Fig1) should be confirmed using *Mex-Gal4* which shows more uniform activation patterns compared to *Myo-Gal4*.

We thank the Reviewer for this suggestion. We used the *Mex-Gal4 tub-Gal80^{ts}* system (*Mex^{ts}*) to examine the effect of EI on ISC division rates on a daily basis and the smurf assay for gut barrier integrity. The results of using the *Mex^{ts}* driver show an alleviation of aged hyperplasia and an improvement in gut barrier function (Extended Data Fig.1a, b), which is comparable to the results of using *Myo1A^{ts}*.

Extended Data Fig. 1a, b

a, Time course of the statistics of PH3⁺ mitotic ISCs per midgut in UT and EI flies. Flies were transferred to 30°C for 12h at 4d to induce a transient EC ablation (*Mex^{ts}>UAS-Reaper*). **b**, Statistics of percentage of smurf flies in UT and EI group. n, number of flies. UT: n=60, 60. EI: n=55, 55.

10. Fig.6g states that new EC improve the SJ Dlg1 staining with old EC. This phrase ('indicating that the new ECs improved the SJs of the old ECs surrounding them.') is very speculative as no expansion or high-resolution microscopy was performed that would show from which of the two cell membranes the stronger signal originates. It could very well be that all additional signal intensity derives from the new EC. For this, in the same samples, SJ of old EC contacting each other could be measured vs mixed vs new-new EC.

We thank the reviewer for raising this issue and for their insightful suggestions. We proposed that the newly generated ECs improve the SJs of the surrounding old ECs. As the reviewer mentioned, we could not tell whether the Dlg1 between new and old came from the new ECs or the old ECs. However, we found that if we focus on these old ECs adjacent to new ECs, then these "boundary old ECs (b old ECs)" not only have edges in contact with new ECs, but also have edges in contact with "non-boundary old ECs (old ECs)" (Fig. 6g). The Dlg1 intensity between these "old-b old" is significantly higher than the Dlg1 intensity between "old-old" (Fig. 6g-h), suggesting that the SJ of the b old ECs was improved by the new ECs. In addition, as suggested by the reviewer, we also compared the Dlg1 staining intensity between "b old-new" and "new-new" ECs, and there was no significant difference in Dlg1 intensity, suggesting that boundary old ECs have similar Dlg1 levels to the new ECs. We have placed these new statistics in Fig. 6h to better illustrate those ideas.

Fig.6 g-h

g, Representative images of the Dlg1 staining in 15d UT and EI midgut. Three types of ECs are present in the EI midgut, RFP⁻ new ECs, RFP⁺ boundary old ECs adjacent to new ECs (b old), and RFP⁺ old ECs not adjacent to new ECs (old). Close-up views show Dlg1 staining in the region of old-old EC contacted (a') and b old-new EC contacted (b'). Cartoon illustrates the observation that new ECs (RFP⁻) restore SJs (green) in b old ECs (RFP⁺). **h**, Statistics on the intensity of Dlg1 staining in four SJs generated between three different ECs in **g**.

11. Functionality of RNAi stocks must be validated. Either by citing papers that validated knockdown efficiency of particular stocks, qPCR or at least reproduction with a second stock targeting the same mRNA.

We thank the Reviewer for raising this question and kindly suggestions. In this manuscript, we used 9 RNAi stocks: (1) *UAS-Atg1 RNAi*, from Bloomington Drosophila Stock Center, BL44034; (2) *UAS-Atg8a RNAi*, BL34340; (3) *UAS-CncC RNAi*, from Tsinghua Fly Center, THU1052; (4) *UAS-Dlg RNAi*, THU1938; (5) *UAS-Cora RNAi*, BL28933; (6) *UAS-Mesh RNAi*, NIG-Fly #12074R-1; (7) *UAS-Tsp2A RNAi*, NIG-Fly #11415R-2; (8) *UAS-Lamin RNAi*, VDRC#45635 and (9) *UAS-LamC RNAi*, TH201500538.S.

1. (1) *UAS-Atg1 RNAi* (BL44034) and (2) *UAS-Atg8a RNAi* (BL34340) were used in ECs to block the autophagy in a Nature paper published in 2021⁵. We have cited this paper in the manuscript. In addition, both RNAi stocks (different genes in the same pathway) give us the same hyperplasia phenotype under iTRF treatment at 70 d (Fig. 7h), indicating both RNAi stocks are validated.

2. (3) *UAS-CncC RNAi* was validated by the DHE staining: A significant increase of ROS levels in ECs was observed after *CncC RNAi* knockdown (Extended Data Fig. 5e).

Extended Data Fig. 5e Representative images of DHE staining in control (*attp2*) and *CncC* knockdown (*Myo1A^{ts}>CncC RNAi*) posterior midgut. ECs were labeled by *Myo1A^{ts}>GFP*.

1. (4) *UAS-Dlg1 RNAi* and (5) *UAS-Cora RNAi* were validated by Dlg1 and Cora antibody staining, respectively. The Dlg1 and Cora staining were absent in ECs following *Myo1A^{ts}>Dlg1 RNAi* or *Cora RNAi* for 6 days (Extended Data Fig. 7g). These experiments also validated the specificity of the antibodies.

Extended Data Fig. 7g, Upper, schematic of the genetic manipulations. Lower, representative images of Dlg1/Cora and PH3 staining in control (*attp2*) and *Myo1A^{ts>}Dlg1/Cora RNAi* posterior midgut. White arrowheads indicate the locations of PH3⁺ cells.

4. Both (6) *UAS-Mesh RNAi* (NIG-fly #12074R-1) and (7) *UAS-Tsp2A RNAi* (NIG-fly #11415R-2) stocks were used by M. Furuse's lab to disrupt the SJs between ECs⁶⁻⁹. We have cited their work in the manuscript. In addition, all four RNAi stocks (Dlg1, Cora, Mesh and Tsp2A) targeting the SJ between ECs showed the same hyperplasia phenotype (Extended Data Fig. 7h, i), validating both the (6) *UAS-Mesh RNAi* and (7) *UAS-Tsp2A RNAi* stocks.

Extended Data Fig. 7h-i

h, Statistics of PH3⁺ cells in midgut of control (*attp2*) and knockdown indicated SJ components in ECs. n=17-22. **i**, Upper, schematic of the method that knocked down SJs after the recovery from EI (bleomycin-fed). Lower, statistics of PH3⁺ cells in midguts at indicated days. n=16-23.

5. (8) *UAS-Lamin RNAi* (VDR#45635) stock was used by Y. Zheng's lab to knock down Lamin in flies¹⁰. We have cited their work in the manuscript. In addition, both (8) *UAS-Lamin RNAi* and (9) *UAS-LamC RNAi* (TH201500538.S.) were validated by Lamin and LamC antibody staining. The Lamin and LamC staining were absent in ECs following *Myo1A^{ts>}Lamin RNAi* or *LamC RNAi* for 6 days (Extended Data Fig. 6d, e). These experiments also validated the specificity of the Lamin and LamC antibodies.

Extended Data Fig. 6d-e

d, Upper, schematic of genetic manipulations. Lower, representative images of Lamin and PH3 staining in control (*attp2*) and *Myo1A^{ts>}Lamin RNAi* posterior midgut. Due to the knockdown of *Lamin* in polyploid ECs, the remaining Lamin staining is present in the diploid cells. Yellow arrowheads indicate the location

of PH3⁺ cells. **e**, Upper, schematic of genetic manipulations. Lower, representative images of LamC staining in control (*atp2*) and *Myo1A^{ts}>LamC RNAi* posterior midgut.

The idea of reverse 'ReDDM' labelling the EC population is really useful (and cool). It also shows that the authors use high quality fly food stimulating low turnover in combination with flipping every 48h (M&M), which allows observation in homeostasis. The authors should also consider discussing that hyperplastic cells remain as their homeostatic midgut turnover is really low.

We thank the Reviewer for their recognition of our methodology and insightful suggestions for the discussion. We added one discussing sentence: "To our surprise, although the homeostatic midgut turnover is low, a significantly high number of PH3⁺ cells (hyperplasia) abruptly appeared at 45d, and the hyperplasia persisted thereafter (Fig. 1f)."

Regarding this method, the cartoons in Fig.3+4 sum up data that is not shown in the confocal images and thus are an assumption. *esg+* are not addressed so it is not formally proven that these guts have hyperplasia. This needs to be shown directly (*esg*-GFP from Leanne Jones lab or armadillo/pros stainings) or removed.

We thank the Reviewer for those suggestions. We have removed the green cells from the cartoon in Fig. 3j, as the Fig. 3h-3j was intended to illustrate the dynamic EC turnover patterns in aged UT and EI flies, which are unrelated to *esg*⁺ cell hyperplasia.

Fig. 3h-j

h, Schematic illustration of the EC labeling and tracing in the aged midgut after the EI. EC labeling is achieved by transient expression of H2B-RFP in ECs using *Myo1A^{ts}>H2B-RFP* for 24h. **i**, **j**, Representative images **i** and cartoon illustrations **j** of tracing of ECs by the transiently labeled H2B-RFP in UT and EI midguts at 45d, 48d and 50d. Newly generated ECs are RFP⁺ and visualized by the large polyploid DAPI staining.

To directly show *esg*⁺ cell hyperplasia in Fig. 4, we followed the reviewer's suggestion by either combining *esg*-GFP with *Mex^{ts}>UAS-H2B-RFP* or staining Arm+Pros in

Myo1A^{ts}>*UAS-H2B-RFP* flies (Extended Data Fig. 4). Using these two methods, we show that *esg*⁺/*Arm*⁺ cell hyperplasia occurred in aged UT midguts (42 d) and immediately after EI midguts (6 d). However, hyperplasia was significantly reduced in aged EI midgut (42 d).

Extended Data Fig. 4 | a, Upper, schematic illustration of the method to track the turnover of puparium labeled H2B-RFP⁺ ECs in UT and EI (bleomycin-fed 24h at 4d) midguts. Lower, representative images of the tracing pattern of H2B-RFP⁺ ECs and *esg*-GFP labeled ISC/EBs in UT and EI midguts at 6d, 10d and 14d. Midgut regions were selected from R4C/R5A. Newly generated ECs (white arrowheads) are RFP⁺ and visualized by the large polyploid DAPI staining. **b**, Cartoon showing the distribution of puparium-

labeled RFP⁺ (red) ECs versus newly generated RFP⁻ (blue) ECs and *esg-GFP⁺* (Green cells) ISC/EBs in the UT and EI midgut during aging. The red color, representing the amount of H2B-RFP in the EC nuclear, fades with age. Green cells are ISC-EB pairs. **c**, Upper, schematic illustration of the second H2B-RFP labeling and tracing of ECs. Lower, representative images of the tracing pattern of RFP⁺ ECs and *esg-GFP⁺* ISC/EBs in UT and EI midguts after 22d. White arrowheads indicate the newly generated RFP⁻ ECs, yellow arrowheads indicate PH3⁺ cells. **d**, Cartoon showing the abrupt collective turnover of RFP⁺ ECs from 37d to 42d, contrasting with the formation of the age mosaic of EI midgut. **e**, Upper, schematic illustration of the method to track the turnover of puparium labeled H2B-RFP⁺ ECs in UT and EI (bleomycin-fed 24h at 4d) midguts. Lower, representative images of the tracing pattern of H2B-RFP⁺ ECs and the ISC/EB marker Armadillo (Arm) and the EE marker Prospero (Pros) staining in UT and EI midguts at 6d, 10d and 14d. **f**, Second H2B-RFP labeling and tracing of ECs. White arrowheads indicate the newly generated RFP⁻ ECs. **g**, Quantification of the number of *esg-GFP⁺* cells in midguts of UT and EI flies (**a** and **c**). Data are mean ± SEM. Significance was determined using two-tailed unpaired t test. Scale bars, 20 μm.

Minor

12. Regarding the data of Fig.7: The authors may consider to reveal the full extent of the EI/iTRF effect with survival curves with ITRF intervention at 60d to reveal whether iTRF is able to extend life span when hyperplasia already happened and 'help' old individuals.

We thank the Reviewer for this insightful suggestion. It has been reported that the timing of Intermittent Fasting (IF) and iTRF regime is important to mediated lifespan extension^{5,11}. TRF/iTRF diet in older flies (days 40-50) did not extend lifespan (Response Fig. 4)⁵. As iTRF is unable to induce age mosaic in the aged hyperplastic midgut, these results are consistent with our hypothesis that iTRF extends lifespan by inducing EC age mosaic during early life. Therefore, we do not expect iTRF intervention in the midgut at 60 d to have a lifespan extending effect.

Response Fig. 4. One previous study in its *Extended Data Fig. 2f* showed that iTRF treatment between days 40 and 50 of adult female flies did not extend lifespan (Ulgherait, Matt et al., 2021; PMID: PMC9395244)⁵.

13. In the same line: Conclusions could be strengthened if Myo> forcing Atg1/8a expression a la EI would extend lifespan.

We thank the reviewer for this insightful suggestion. We agree with the reviewer that it would be informative to test whether *Myo1A^{ts}>Atg1/8a* could mimic an EI effect to extend lifespan. Indeed, recent work has confirmed that early *Atg1* overexpression in ECs extends lifespan and prevents age-related loss of intestinal integrity (Response Fig. 5)¹². These findings support our conclusions.

Response Fig. 5. One previous study in its **Fig. 5b, c** showed that chronic and day 1–15 overexpression of *Atg1* specifically in enterocytes extended lifespan to the same degree as rapamycin (Juricic, Paula et al., 2022; PMID: PMC10154223)¹².

14. For gene description/protein labelling please refer to Flybase nomenclature (e.g. Dlg1 instead of Dlg) throughout the manuscript.

We thank the Reviewer and have made this correction.

15. It should also be stated clearly where in the midgut these measurements were performed. Regions differ vastly and investigations should only focus on one region such as R5.

We thank the Reviewer for pointing this out. All PH3⁺ cell counts were performed throughout the midgut. All representative images were taken in regions of R4C and R5A. We have added this information in the legends of Fig. 1i, Fig. 4a, Extend Data Fig. 4a and in the Materials and Methods.

16. The role of Lamins in the Drosophila midgut was investigated by Joerg Großhans lab before and should be cited accordingly.

We thank the Reviewer for reminding us. We have cited the following two references accordingly in our manuscript:

Petrovsky, R. and J. Grosshans, *Expression of lamina proteins Lamin and Kugelkern suppresses stem cell proliferation*. Nucleus, 2018. **9**(1): p. 104-118.

Petrovsky, R., G. Krohne, and J. Grosshans, *Overexpression of the lamina proteins Lamin and Kugelkern induces specific ultrastructural alterations in the morphology of the nuclear envelope of intestinal stem cells and enterocytes*. *Eur J Cell Biol*, 2018. **97**(2): p. 102-113.

We again thank the Reviewer for their insightful comments and suggestions!

Reference:

- 1 Antonello, Z. A., Reiff, T., Ballesta-Illan, E. & Dominguez, M. Robust intestinal homeostasis relies on cellular plasticity in enteroblasts mediated by miR-8-Escargot switch. *The EMBO journal* **34**, 2025-2041, doi:10.15252/embj.201591517 (2015).
- 2 Zipper, L., Corominas-Murtra, B. & Reiff, T. Steroid hormone-induced wingless ligands tune female intestinal size in *Drosophila*. *Nature communications* **16**, 436, doi:10.1038/s41467-024-55664-2 (2025).
- 3 Ahmed, S. M. H. *et al.* Fitness trade-offs incurred by ovary-to-gut steroid signalling in *Drosophila*. *Nature* **584**, 415-419, doi:10.1038/s41586-020-2462-y (2020).
- 4 Zipper, L., Jassmann, D., Burgmer, S., Gorlich, B. & Reiff, T. Ecdysone steroid hormone remote controls intestinal stem cell fate decisions via the PPARgamma-homolog Eip75B in *Drosophila*. *Elife* **9**, doi:10.7554/eLife.55795 (2020).
- 5 Ulgherait, M. *et al.* Circadian autophagy drives iTRF-mediated longevity. *Nature* **598**, 353-358, doi:10.1038/s41586-021-03934-0 (2021).
- 6 Izumi, Y., Furuse, K. & Furuse, M. Septate junctions regulate gut homeostasis through regulation of stem cell proliferation and enterocyte behavior in *Drosophila*. *Journal of cell science* **132**, doi:10.1242/jcs.232108 (2019).
- 7 Izumi, Y., Motoishi, M., Furuse, K. & Furuse, M. A tetraspanin regulates septate junction formation in *Drosophila* midgut. *Journal of cell science* **129**, 1155-1164, doi:10.1242/jcs.180448 (2016).
- 8 Yanagihashi, Y. *et al.* Snakeskin, a membrane protein associated with smooth septate junctions, is required for intestinal barrier function in *Drosophila*. *Journal of cell science* **125**, 1980-1990, doi:10.1242/jcs.096800 (2012).
- 9 Izumi, Y., Yanagihashi, Y. & Furuse, M. A novel protein complex, Mesh-Ssk, is required for septate junction formation in the *Drosophila* midgut. *Journal of cell science* **125**, 4923-4933, doi:10.1242/jcs.112243 (2012).
- 10 Chen, H., Zheng, X. & Zheng, Y. Age-Associated Loss of Lamin-B Leads to Systemic Inflammation and Gut Hyperplasia. *Cell* **159**, 829-843, doi:10.1016/j.cell.2014.10.028 (2014).
- 11 Catterson, J. H. *et al.* Short-Term, Intermittent Fasting Induces Long-Lasting Gut Health and TOR-Independent Lifespan Extension. *Current biology : CB* **28**, 1714-1724 e1714, doi:10.1016/j.cub.2018.04.015 (2018).
- 12 Juricic, P. *et al.* Long-lasting geroprotection from brief rapamycin treatment in early adulthood by persistently increased intestinal autophagy. *Nature Aging* **2**, 824-836, doi:10.1038/s43587-022-00278-w (2022).

Reviewer #2 (Remarks to the Author):

In this article titled "Age mosaic of gut epithelial cells prevents aging", the authors have explored the idea of healthy aging by inducing early gut epithelial renewal in young organisms. They have performed extensive transcriptomic analyses of aging *Drosophila* midguts. This data set supports the idea that early injuries to epithelial lining is beneficiary to rejuvenate the gut, thereby increasing the lifespan. Even though the idea of early injury seems interesting, this reviewer is not completely sure how relevant it is in a model like mammalian gut which undergoes a turnover every 3-4 days for several years whereas *Drosophila* has a turnover of 2-3 weeks for few months only. The generation of new ECs by TA cells surrounding the villi is by itself providing this event. This reviewer thinks that their idea of pushing this approach towards therapeutic purpose could be a bit of an overstatement.

We thank the Reviewer for the positive assessment of our work and for appreciating our efforts in performing transcriptomic analyses of the ageing midgut. We fully agree with the reviewer's comparison of the *Drosophila* gut with the mammalian gut and share the view that the mammalian gut epithelium may not be as age mosaic as the *Drosophila* gut epithelium. However, the Cell Metabolism paper from Martin Hetzer's lab¹ shows that even the liver, which has a high turnover rate, contains cells that are as old as the animal, forming an age mosaic of liver cells. In addition to the liver, the age mosaic is also present in the central nervous system (CNS) and pancreas of mice. Our elucidation of the age mosaic in *Drosophila* may shed light on the physiological role of the age mosaic in mammals.

Major Comments:

1. The main idea that authors propose is that in aged guts, old enterocytes (ECs) undergo collective turnover, leading to hyperplasia. When they say that ECs undergo collective turnover, do they mean that ECs are lost all at once or still lost in a controlled fashion? Can the authors look for markers for cell death in ECs of aged midgut to get an idea of the dynamics of epithelial removal?

We thank the Reviewer for raising this interesting question. In our lineage tracing of H2B-RFP⁺ ECs experiments, we can observe a drastic reduction of ECs within a few days (Fig. 4). We believe that collective EC turnover is a process that occurs within 24 hours, rather than an "at once" loss that occurs in minutes or hours. Following the reviewer's suggestion, we observed scattered EC deaths in 42d at 18°C midguts by TUNEL staining, but did not find aggregated or clustered EC apoptosis (Response Fig. 6), suggesting that the ECs lost are not in a controlled fashion. Our data (Fig. 5) suggest that the accumulating ROS levels in ECs reached the threshold triggering cell death in the majority of ECs during midgut age, resulting in 'collective EC turnover'.

Response Fig. 6. TUNEL staining in 42d UT midgut. Scale bars, 20 μ m.

The strategy they use for inducing early injury (EI) is by either genetic means by expressing Reaper in ECs or using pharmacological stressors. These treatments can also induce an uncontrolled collective turnover of ECs but seem not to induce hyperplasia when the guts are left to recover.

Does this mean that the way a midgut recovers depends also on the age of the organism? What is the extent of damage to ECs by transient expression of Reaper, bleomycin, and paraquat?

Thank you for raising these important questions. The strategies we used for inducing EI did not induce the collective EC turnover that occurs in the aged midgut. The data in Fig. 4 show that RFP⁺ old EC in the UT gut still accounted for more than 60% of the EC at 37 d (Fig. 4a, e), but collectively disappeared by 42 d (Fig. 4a, e). In contrast, both bleomycin feeding and transient *Reaper* expression induced EC turnover was progressive (Fig. 4a, e, g and h), creating an age mosaic of EC in the gut. We therefore believe that EI's recovery is not due to its young age, but rather to the fact that collective turnover is not generated.

Fig. 4a, e, h, g

a, Upper, schematic illustration of the method to track the turnover of puparium labeled H2B-RFP⁺ ECs in UT and EI (bleomycin-fed 24h at 4d) midguts. Lower, representative images of the tracing pattern of H2B-RFP⁺ ECs in UT and EI midguts at 6d, 10d, 14d, 37d and 42d. Midgut regions were selected from

R4C/R5A. Newly generated ECs (white arrowheads) are RFP⁻ and visualized by the large polyploid DAPI staining. **e**, Statistics of the ratio of RFP⁺ ECs to total ECs. **g, h**, Statistics of the ratio of RFP⁺ ECs to total ECs **g** and representative images of the tracing pattern of H2B-RFP⁺ ECs in UT and *Myo1A^{ts}>Reaper* midguts **h**. **n**, numbers of regions, n=10-27 per group. White arrowheads indicate occasional newly formed ECs in the UT midgut.

The way pharmacological treatments (bleomycin or paraquat) are used in this study on their own has a toxic effect. Could the authors optimize the exposure time and dosage of these treatments so that it is enough to generate a turnover of epithelial cells but won't have an overall toxic effect?

We thank the reviewer for this insightful suggestion. We have tried many combinations of dosage and feeding time, and the current feeding conditions have been our optimal choice. In our search, we have found that feeding for less than 24 hours causes variation in food intake between *Drosophila* individuals, leading to instability in the gut phenotype. Based on a 24-hour feeding period, the current doses of bleomycin and paraquat are already the lowest concentrations that can produce a stable phenotype. Higher concentrations would result in toxicity leading to significant mortality in flies.

2. This reviewer finds it difficult to understand why flies that are of different ages are classified as old. For example, in Figure 1B, there is data from flies that are 100 and 150 days old whereas in Figure 1C or 1I, there is data from only till 40 or 21 days. Are these flies old or are they stressed? Have the authors looked for some markers of aging to be sure that these are old flies? Is this difference due to the treatment itself or genetic background? A survivorship curve for these conditions like in Figure 1A would help the reader understand the choice of days of age for classifying as old flies.

The difference in age is due to the temperature at which the experiment was carried out. As the fly is a poikilotherm, the temperature has a significant effect on its lifespan² (Response Fig. 7). At 18°C (Fig. 1a and 1b), fly can survive for over 100 days. At 25°C (Fig. 1c and Extended Data 6a, b), flies begin to die after 40 days.

Response Fig. 7. One previous study in its **Fig.3A** showed that effects of temperature on adult lifespan (Linford, Nancy J et al., 2013; PMID: PMC3582515)².

3. In Figure 3, the authors conclude that EI leads to a slowdown of EC turnover in the aged midgut. It is known that ISC hyperproliferation can accelerate EC turnover and aged UT midguts have hyperplasia. In contrast, EI midguts do not exhibit hyperplasia. Could the accelerated EC turnover in aged UT midguts be an outcome of hyperplasia?

Thanks for raising this interesting question. Yes, it has been demonstrated that when hyperplasia happens in the aged UT midgut, the constant production of EBs by ISCs crowds the space where ECs contact the basement membrane³⁻⁵, causing a constant EC turnover⁶. Therefore, we believe that once hyperplasia has formed, it creates a vicious cycle of constant EC turnover.

Can the authors block ISC proliferation and follow EC turnover in the aged UT midguts?

We thank the Reviewer for this insightful suggestion. It has been shown that once ISC proliferation is blocked, EC turnover stops⁷. In addition, when we temporarily halted (TH) ISC mitosis of aged hyperplastic midguts (65d at 18°C), we found that hyperplasia was alleviated in TH-treated midgut up to 95d (Fig. 6k, m and n), indicating the EC turnover stopped after blocking the ISC proliferation.

Fig. 6k, m, n.

k, Upper, schematic of temporarily halting (TH) the ISC division in 65d hyperplastic midgut by *esg^{ts}>Cdk2 RNAi* for 7 days. **m**, **n**, Representative images **m** and statistics **n** of PH3⁺ cells in aged control and TH midgut. n=14-18.

4. Authors propose the idea of accumulation of ROS in old ECs as an upstream signal for

collective EC turnover. Can the authors test this directly by reducing the intracellular ROS maybe by feeding the flies with ROS scavenger like N-Acetyl Cysteine and then check EC turnover in UT flies upon aging?

We thank the Reviewer for this insightful suggestion. Following the reviewer's suggestion, we fed the flies with the ROS scavenger N-Acetyl Cysteine (NAC) to remove intracellular ROS in the midgut. The DHE staining was significantly reduced after NAC feeding (Extended Data Fig. 5i), demonstrating the relief of ROS in the midgut epithelium. In line with our hypothesis, the NAC-feeding group of flies did not develop hyperplasia at 23 d at 25°C (Extended Data Fig. 5j), suggesting that blocking the accumulation of ROS prevents collective EC turnover.

Extended Data Fig. 5i, j

i, Representative images of DHE staining of live midgut of 21d UT flies and 21d flies treated with NAC (DHE, red; GFP indicated ISCs/EBs). j, Statistics of PH3⁺ mitotic ISCs per midgut in UT- and NAC-treated flies at 23 days. Data are mean ± SEM. Significance was determined using two-tailed unpaired t test. Scale bars, 20 μm.

Can the authors also check the EC turnover using their technique upon knocking down CncC to increase cellular ROS?

We thank the Reviewer for this insightful suggestion. Following the reviewer's suggestion, we used the *Mex^{ts}>UAS-H2B-RFP* system to knock down *CncC* in *H2B-RFP* labelled ECs. While *H2B-RFP* labelled ECs in UT flies did not turnover significantly at 20 day at 18°C, *H2B-RFP* labelled *CncC* knockdown ECs lost rapidly over the following days at 18°C, demonstrating that elevated ROS levels promote EC turnover (Extended Data Fig. 5g, h).

Extended Data Fig. 5g, h.

g, Upper, schematic of the H2B-RFP EC labeling and tracing in control (*Mex^{ts}>UAS-H2B-RFP/attp2*) and *CncC* knockdown (*Mex^{ts}>UAS-H2B-RFP/UAS-CncC RNAi*) flies, flies were transferred to 30°C for 6 days at 4d to knock down *CncC* in ECs and labelled ECs. Lower, representative images of the RFP tracing pattern in midguts of 10d, 15d and 20d flies. **h**, Statistics of the ratio of RFP⁺ ECs to total ECs in selected areas of control and *CncC* knock down flies during aging. Data are mean ± SEM. Significance was determined using two-tailed unpaired t test. Scale bars, 20 µm.

Interestingly, from Figure 5d, it is not clear if all old ECs in 30D Ut midguts are in high ROS state. Can the authors plot the mean intensity of GFP in RFP⁺ ECs between 15D UT and 30D UT conditions?

Sure. We plotted the mean intensity of *gstD1*-GFP in RFP⁺ ECs among 15d, 20d and 30d UT conditions (Extended Data Fig. 5a).

Extended Data Fig. 5a, Statistics of *gstD1*-GFP intensity in RFP⁺ ECs of 15d, 20d and 30d flies during aging. n, numbers of ECs. 10-12 midguts per group.

5. Is inhibition of collective EC turnover sufficient to inhibit hyperplasia?

Yes, the EI treatment, which creates an age mosaic of ECs, prevents the collective turnover of ECs, resulting in the alleviation of hyperplasia.

Can the authors express a blocker of cell death like p35 in ECs to see if this is sufficient to block hyperplasia?

We thank the Reviewer for this insightful suggestion. Ectopic expression of *p35/Diap1* in aged midgut ECs cannot block hyperplasia (Response Fig. 8). However, it has been demonstrated that apoptosis does not play a major role in EC turnover⁸. There are at least two other mechanisms: (1) EC cell shedding (apical extrusion) upon bacterial infection⁹, and (2) a new apoptosis-independent cell death mechanism, called erebosis, mediates EC turnover during midgut epithelial homeostasis⁸. Therefore, although we believe that collective EC turnover induces hyperplasia, we don't have a way to block all EC turnover to see if this is sufficient to block hyperplasia.

Response Fig. 8. Upper, schematic of temporarily halting (TH) the EC apoptosis in 60d hyperplastic midgut by *Myo1A^{ts}>UAS-p35/Diap1* for 7 days. Lower, statistics of PH3+ cells in aged control and TH midgut.

6. The authors suggest that the septate junctions (SJs) in old ECs are rescued when they are neighboring a new EC (see model 6g). But in Figure 6b or e, SJs are well restored in old ECs of EI midguts irrespective of whether they neighbor a new EC or old EC.

The RFP+ old ECs in Figure 6b or 6e are all in contact with the RFP- new ECs (at least on one side), resulting in their SJs being well restored.

If disrupted SJs are the driving force for collective EC turnover, I am puzzled by finding large clones of old ECs with disrupted SJs in 15D EI midguts. Can the authors test if restoring the expression of SJ components in old ECs is sufficient to block collective EC turnover and hyperplasia?

We thank the Reviewer for this insightful suggestion. By temporarily halted the ISC overproliferation at 65d at 18°C, we restored the SJ components Dlg1 and Cora's staining between ECs even at 85d (Fig. 6j-l and Extended Data Fig. 7j, k). Coordinately, the hyperplasia was blocked in old midguts, suggesting the collective EC turnover was also blocked (Fig. 6n).

Fig. 6j, k, l.

j, A cartoon shows the hypothesis that halting the hyperplasia restores the correct SJs among ECs. **k**, Upper, schematic of temporarily halting (TH) the ISC division in 65d hyperplastic midgut by *esg^{ts}>Cdk2 RNAi* for 7 days. Lower, representative images of Dlg1 staining in control and TH midgut at 85d. **l**, Statistics of Dlg1 staining in 85d control and TH midgut.

Extended Data Fig. 7j, k, Representative images **j** and statistics **k** of the Cora staining in control and TH midgut at 85d.

Fig. 6n, Statistics of PH3⁺ cells in aged control and TH midgut.

During the revision, we failed to obtain the overexpression stocks of SJ components (*UAS-Dlg1*, *UAS-Cora*, etc.), so we cannot directly test whether overexpression of these SJ components in aged flies could block EC turnover. However, previous studies have shown that overexpression of Snakeskin (*Ssk*, an important SJ component) in ECs significantly improves barrier integrity, alleviates hyperplasia during aging, and extends lifespan¹⁰. Therefore, we believe that restoring SJs in old fly ECs could block collective EC turnover and prevent hyperplasia.

Minor Comments:

1. How does the Lamin distribution in the newly produced ECs in 50D UT midgut look like?

As ECs in 50d UT midgut are all newly generated (RFP⁻), most of the ECs have nuclear-membrane localized Lamin staining (Response Fig. 9).

Myo1A^{ts}>UAS-H2B-RFP @18°C

Response Fig. 9. Representative images of Lamin staining in newly generated ECs (RFP⁻) in the 50d UT aged midgut. Scale bars, 20 μm.

Is restoring Lamin sufficient in old ECs to prevent their collective turnover?

We thank the reviewer for this insightful suggestion. We transiently overexpressed Lamin in old RFP-labeled ECs at 25d, in which Lamin expression and localization were confirmed by antibody staining at 27d (Response Fig. 10a, b). However, compared to the control group (*Mex^{ts}>UAS-H2B-RFP* crossed with *attp2*), transiently overexpression of Lamin in ECs did not prevent their collective turnover (Response Fig. 10b-d).

Since overexpression of Lamin induces specific ultrastructure alterations in the nuclear envelope of ECs¹¹ and suppresses ISC proliferation¹², prevention of EC turnover may require physiological Lamin expression. Alternatively, Lamin expression itself may not be sufficient to prevent the collective turnover of ECs.

Response Fig. 10a-d

a, schematic of the H2B-RFP to label and trace EC in control (*Mex^{ts}>UAS-H2B-RFP/attp2*) and Lamin overexpression (*Mex^{ts}>UAS-H2B-RFP/UAS-Lamin*) flies, flies were transferred to 30°C for 2 days at 25d to overexpress Lamin in ECs and labelled ECs. **b**, Representative images of Lamin staining in RFP⁺ ECs in the midguts of 27d control (*attp2*) and Lamin-overexpression flies. **c**, Representative images of the RFP tracing pattern in midguts of control (*attp2*) and Lamin-overexpression flies at 32d and 40d. **d**, Statistics of the ratio of RFP⁺ ECs to total ECs in selected areas of control and Lamin overexpressed flies during aging. Data are mean ± SEM. Significance was determined using two-tailed unpaired t test. Scale bars, 20 μm. 15-20 midguts per group.

2. Similarly, in old UT midguts (30D for example), there is still a bit of EC turnover. When these new ECs form in between old ECs, does this rescue the septate junctions (SJs) in old ECs?

We thank the reviewer for this interesting question. We have added new representative images of Dlg1 staining with RFP tracing of old ECs in the 20d UT midgut, where RFP⁻ new ECs are occasionally present in the gut epithelium. In these images, new ECs not only improved Dlg1 staining at their own cell borders, but also significantly improved Dlg1 staining of old ECs surrounding them (Response Fig. 11a). Therefore, we divided the RFP⁺ old EC into two categories, one for old EC in contact with RFP⁻ new EC, termed "old EC with contact new", and the second for old EC without any contact with RFP⁻ new EC, termed "old EC without contact new". The statistics showed that "old EC with contact new" had a significantly higher Dlg1 staining intensity than "old EC without contact new" (Response Fig. 11b), suggesting the new ECs rescued the SJs in their neighboring old ECs.

Response Fig. 11a, b

a, upper, the schematic shows the method for monitoring Dlg1 in the midgut with puparium H2B-RFP labeled ECs. lower, representative images of the Dlg1 staining in 20d UT midgut, where RFP⁻ new ECs are occasionally present in the gut epithelium. Two types of RFP⁺ old EC in the UT midgut, one for old EC in contact with RFP⁻ new EC, termed "old EC with contact new", and the second for old EC without any contact with RFP⁻ new EC, termed "old EC without contact new". Yellow arrows indicate the SJs of old ECs that in contact with RFP⁻ new ECs. **b**, Statistics on the intensity of Dlg1 staining in "old EC with contact new" and in "old EC without contact new" in **a**. Data are mean \pm SEM. Significance was determined using two-tailed unpaired t test. Scale bars, 20 μ m.

3. In general, it would be great if the confocal micrographs are not too cropped for example, like in Figure 5d, g, 6b etc. I prefer to see a bigger overview of the midgut region so that the mosaicity of old and new ECs is clearer.

We have added new representative bigger overview of the midgut region in Extended Data Fig.5b for Figure 5d, Extended Data Fig.6a for Figure 5g, and Extended Data Fig. 7b for Figure 6b.

b *Myo1A^{ts}>UAS-H2B-RFP/gstD1-GFP@18°C*

Extended Data Fig. 5b, Representative images of *gstD1-GFP* staining in 30d UT and EI (bleomycin-fed 24h at 4d) midgut. Scale bars, 20 μm.

Extended Data Fig. 6a, Representative images of Lamin staining in the UT and EI (bleomycin-fed 24h at 4d) midgut. Scale bars, 20 μ m.

Extended Data Fig. 7a, b

a, The schematic shows the method for monitoring Dlg1 in the midgut with puparium H2B-RFP labeled ECs. **b**, Representative images of Dlg1 staining in the UT at 5d and 30d, and in the EI midgut at 30d. Scale bars, 20 μ m.

We again thank the Reviewer for their comments and insightful suggestions!

Reviewer #3 (Remarks to the Author):

We thank all the Reviewers for their positive comments and constructive suggestions on our work.

Reference:

- 1 Arrojo e Drigo, R. *et al.* Age Mosaicism across Multiple Scales in Adult Tissues. *Cell metabolism* **30**, 343-351.e343, doi:10.1016/j.cmet.2019.05.010 (2019).
- 2 Linford, N. J., Bilgir, C., Ro, J. & Pletcher, S. D. Measurement of lifespan in *Drosophila melanogaster*. *Journal of visualized experiments : JoVE*, doi:10.3791/50068 (2013).
- 3 Rodriguez-Fernandez, I. A., Tauc, H. M. & Jasper, H. Hallmarks of aging *Drosophila* intestinal stem cells. *Mechanisms of ageing and development* **190**, 111285,

- doi:10.1016/j.mad.2020.111285 (2020).
- 4 Jasper, H. Intestinal Stem Cell Aging: Origins and Interventions. *Annual Review of Physiology, Vol 82* **82**, 203-226, doi:10.1146/annurev-physiol-021119-034359 (2020).
- 5 Biteau, B., Hochmuth, C. E. & Jasper, H. JNK activity in somatic stem cells causes loss of tissue homeostasis in the aging *Drosophila* gut. *Cell stem cell* **3**, 442-455, doi:10.1016/j.stem.2008.07.024 (2008).
- 6 Patel, P. H., Dutta, D. & Edgar, B. A. Niche appropriation by *Drosophila* intestinal stem cell tumours. *Nature cell biology* **17**, 1182-1192, doi:10.1038/ncb3214 (2015).
- 7 Jin, Y. *et al.* Intestinal Stem Cell Pool Regulation in *Drosophila*. *Stem cell reports* **8**, 1479-1487, doi:10.1016/j.stemcr.2017.04.002 (2017).
- 8 Ciesielski, H. M. *et al.* Erebosis, a new cell death mechanism during homeostatic turnover of gut enterocytes. *PLoS biology* **20**, e3001586, doi:10.1371/journal.pbio.3001586 (2022).
- 9 Zhai, Z., Boquete, J. P. & Lemaitre, B. Cell-Specific Imd-NF-kappaB Responses Enable Simultaneous Antibacterial Immunity and Intestinal Epithelial Cell Shedding upon Bacterial Infection. *Immunity* **48**, 897-910 e897, doi:10.1016/j.immuni.2018.04.010 (2018).
- 10 Salazar, A. M. *et al.* Intestinal Snakeskin Limits Microbial Dysbiosis during Aging and Promotes Longevity. *iScience* **9**, 229-243, doi:10.1016/j.isci.2018.10.022 (2018).
- 11 Petrovsky, R., Krohne, G. & Grosshans, J. Overexpression of the lamina proteins Lamin and Kugelkern induces specific ultrastructural alterations in the morphology of the nuclear envelope of intestinal stem cells and enterocytes. *Eur J Cell Biol* **97**, 102-113, doi:10.1016/j.ejcb.2018.01.002 (2018).
- 12 Petrovsky, R. & Grosshans, J. Expression of lamina proteins Lamin and Kugelkern suppresses stem cell proliferation. *Nucleus* **9**, 104-118, doi:10.1080/19491034.2017.1412028 (2018).

REVIEWERS' COMMENTS

Reviewer #1 (Remarks to the Author):

Dear authors,

during the revision process, the authors addressed all my (and the other reviewers) concerns extensively and significantly improved the manuscript, which I hope to read in NComms soon.

Finally, I want to congratulate the authors for their findings and will be happy to review more of their work in the future.

We would like to thank Reviewer #1 for their kind words and for their constructive suggestions for the improvement.

Reviewer #2 (Remarks to the Author):

I think that the authors have adequately addressed my comments in the revised version of the manuscript. Therefore, I have no further comments.

We would like to thank Reviewer #2 for their constructive suggestions for the improvement.

Reviewer #3 (Remarks to the Author):

We would like to thank Reviewer #3 for their constructive suggestions for the improvement.